# The muscle specific MEF2Dα2 isoform promotes muscle ketolysis and running capacity in mice

Sushil Kumar [ID] [1,7], Xuan Ji[2], Hina Iqbal [ID] [1,3], Xiangnan Guan[4,5], Brittany Mis[1,3,8], Devanshi Dave [ID] [6], Suresh Kumar [ID] [1], Jacob Besler[1], Ranjan Dash [ID] [6✉], Zheng Xia [ID] [4,5✉] & Ravi K Singh [ID] [1,2,3✉]

## Abstract

During prolonged starvation and exhaustive exercise, when there is low availability of carbohydrates, the liver breaks down fatty acids to generate ketone bodies, which are utilized by peripheral tissues as an alternative fuel source. The transcription factor MEF2D undergoes regulated alternative splicing in the postnatal period to produce a highly conserved, muscle specific MEF2Dα2 protein isoform. Here, we discover that compared to WT mice, MEF2Dα2 exon knockout (Eko) mice display reduced running capacity and muscle expression of all three ketolytic enzymes: BDH1, OXCT1, and ACAT1. MEF2Dα2 Eko mice consistently show increased blood ketone body levels in a tolerance test, after exercise, and when fed a ketogenic diet. Lastly, using mitochondria isolated from skeletal muscle, Eko mice show reduced ketone body utilization compared to WT mice. Collectively, our findings identify a new role for the MEF2Dα2 protein isoform in regulating skeletal muscle ketone body oxidation, exercise capacity, and systemic ketone body levels.

**Keywords** MExercise Metabolism; Alternative Splicing; Ketone Body; MEF2 Transcription Factors
**Subject Categories** Metabolism; Musculoskeletal System

## Introduction

Skeletal muscles are essential for movement, postural stabilization, and breathing. In adulthood, human skeletal muscle makes up to 40% of the body weight and is the primary site for insulin-stimulated glucose disposal (Baron et al, 1988; Frontera and Ochala, 2015; Meyer et al, 2002). There is a growing recognition of the role skeletal muscle plays in whole-body metabolism as early disruptions in skeletal muscle carbohydrate and fatty acid metabolism correlate with developing type II diabetes and obesity (DeFronzo and Tripathy, 2009; Holloway et al, 2009; Kim et al,

2000). Conversely, exercise-mediated improved substrate storage and utilization in the skeletal muscle enhances whole-body metabolism to prevent and better manage metabolic diseases and aging-associated decline in bodily function while boosting the overall quality of life (Arad et al, 2020; Colberg et al, 2010; Kelley and Goodpaster, 2001; Stanford and Goodyear, 2014). Thus, skeletal muscle plays a broader role in maintaining whole-body metabolic and energy homeostasis and mediates interorgan cross-talk during normal physiology, aging, starvation, and disease states (Argiles et al, 2016; Mika et al, 2019; Wolfe, 2006).

Alternative splicing is one of the primary mechanisms for expanding the proteomic capacity of the mammalian genome (Liu et al, 2017; Nilsen and Graveley, 2010). The majority of alternative splicing occurs in a tissue-specific manner, including the brain, heart, and skeletal muscle, suggesting that the resultant protein isoforms promote tissue-specific functions (Kalsotra and Cooper, 2011; Pan et al, 2008; Wang et al, 2008). Additionally, the altered function of RNA-binding proteins (RBPs) that regulate alternative splicing in skeletal muscle has been associated with muscle dystrophies in humans (Pistoni et al, 2010; Singh and Cooper, 2012). Similarly, the altered expression of several RBPs which regulate alternative splicing in mouse skeletal muscle, including RBFOX, MBNL, and CELF proteins, causes changes in muscle mass, physiology, and metabolism (Lee et al, 2013; Sellier et al, 2018; Shi and Grifone, 2021; Singh et al, 2018; Ward et al, 2010). Furthermore, hundreds of skeletal muscle transcripts are alternatively spliced during the early postnatal period suggesting a role for the resultant alternate protein isoforms in adult skeletal muscle function and metabolism (Brinegar et al, 2017). However, the function of very few muscle-specific protein isoforms that are generated by alternative splicing has been investigated in detail (Giudice et al, 2016; Nakka et al, 2018; Vecellio Reane et al, 2016).

The myocyte enhancer factor 2 (MEF2) family of transcription factors cooperates with other transcriptional co-regulators and responds to multiple signals to regulate muscle gene expression and differentiation (Molkentin et al, 1995; Molkentin and Olson, 1996; Potthoff and Olson, 2007). Of the four vertebrate *Mef2* genes, *Mef2a*, *Mef2c*, and *Mef2d* are expressed in the adult skeletal muscle (Potthoff and Olson, 2007). MEF2 proteins bind to the A/T-rich

[1]Division of Pediatric Pathology, Department of Pathology, Milwaukee, WI 55226, USA. [2]Department of Pharmacological and Pharmaceutical Sciences, and the Institute of Muscle Biology and Cachexia, College of Pharmacy, University of Houston, Houston, TX 77204, USA. [3]Cardiovascular Center, Medical College of Wisconsin, Milwaukee, WI 55226, USA. [4]Department of Molecular Microbiology and Immunology, Portland, OR 97239, USA. [5]Department of Biomedical Engineering, Oregon Health & Science University, Portland, OR 97239, USA. [6]Department of Biomedical Engineering, Department of Physiology, Medical College of Wisconsin, Milwaukee, WI 55226, USA. [7]Present address: Department of Cell Biology, Neurobiology and Anatomy, Medical College of Wisconsin, Milwaukee, WI 53226, USA. [8]Present address: Department of Medicine, Medical College of Wisconsin, Milwaukee, WI 53226, USA. ✉E-mail: rdash@mcw.edu; xiaz@ohsu.edu; rksingh4@central.uh.edu

consensus sequence through the conserved MADS-MEF2 domain at their N-terminus, while their C-terminal domain is highly divergent and generates diverse protein isoforms through alternative splicing (Martin et al, 1994; McDermott et al, 1993; Yu et al, 1992; Zhu et al, 2005). While the role of MEF2 paralogs in skeletal muscle development has been investigated (Anderson et al, 2015; Potthoff et al, 2007a; Potthoff et al, 2007b), the function of alternative splicing-generated MEF2 protein isoforms in adult skeletal muscle is still not well understood.

The alternative splicing of mutually exclusive α1 or α2 exon in the *Mef2d* transcripts produces the ubiquitous MEF2Dα1 or the muscle-specific MEF2Dα2 protein isoform (Martin et al, 1994; Sebastian et al, 2013). The α2-exon is highly conserved from fish to mammals suggesting an evolutionarily conserved role of the resulting MEF2Dα2 protein isoform in skeletal muscle. The muscle-specific inclusion of the α2-exon is regulated by RBFOX and MBNL splicing factors, and the inclusion of this exon is reduced in the muscles of myotonic dystrophy patients (Singh et al, 2018; Thomas et al, 2017). Previous studies have reported that the Mef2dα2 isoform promotes myoblast fusion in culture and improves muscle regeneration when overexpressed (Sebastian et al, 2013; Singh et al, 2014). However, the in vivo role of MEF2Dα2 protein isoform in adult skeletal muscle has not been studied.

In the present study, we deleted the α2-exon of Mef2d using CRISPR-Cas9 to identify its role in skeletal muscle. Compared to age-matched wild-type (WT) mice, Mef2dα2 exon knockout or Mef2dα2 Eko mice had no changes in overall muscle mass, fiber type, fiber number, glucose tolerance test, and grip strength. However, the Mef2dα2 Eko mice displayed ~30% reduced endurance running capacity compared to age-matched WT mice. In addition, RNA-sequencing of muscle RNA identified modest gene expression changes with 74 upregulated and 40 downregulated genes in sedentary Mef2dα2 Eko compared to sedentary WT mice (adjusted *P* value <0.05). One of the downregulated genes, 3-Hydroxybutyrate Dehydrogenase 1 (Bdh1), encodes a key enzyme in the oxidation of ketone bodies. A targeted analysis of other ketolytic genes also identified reduced expression of OXCT1 and ACAT1. Compared to WT mice and consistent with decreased expression of all ketolytic enzymes in skeletal muscle, Mef2dα2 Eko mice displayed reduced clearance of 3-hydroxybutyrate in a tolerance test and increased blood ketone body availability after exercise and feeding of a ketogenic diet. Thus, our results demonstrate the role of muscle-specific MEF2Dα2 protein isoform in ketone body utilization in postnatal skeletal muscle and maintenance of running capacity in mice.

# Results

## Deleting the Mef2dα2 exon does not affect the splicing of other MEF2 paralogs

The inclusion of α2-exon increases postnatally to >70%, such that MEF2Dα2 is the predominant protein isoform in adult skeletal muscle (Fig. EV1A) (Brinegar et al, 2017). To determine the in vivo role of the muscle-specific MEF2Dα2 protein isoform, we used CRISPR-Cas9 to generate mice lacking the α2 exon. A total of 24 founder mice were generated, and of these, at least three mice had germline transmission of the α2 deletion. We randomly selected two of these founder lines for this study (line 1 and 2 corresponding to tag numbers #1047 and #1056,

respectively). Before phenotyping, both lines were backcrossed to wild-type (WT) mice in the same background (C57BL6/J) for >6 generations. For phenotyping, heterozygous mice (Mef2dα2$^{e-/wt}$ x Mef2dα2$^{e-/wt}$) were crossed to obtain age-, sex-, and genetic background-matched control WT (Mef2dα2$^{wt/wt}$) and Mef2dα2 exon knockout or Mef2dα2 Eko mice (Mef2dα2$^{e-/e-}$). RT-PCR analysis of *Mef2d* transcripts in gastrocnemius and soleus muscles showed complete skipping of the α2 exon in Mef2dα2 Eko mice, whereas muscles from age-matched WT mice showed ~70% inclusion of the α2 exon (Figs. 1A and EV1B). Moreover, the absence of α2 exon in Mef2dα2 Eko mice muscles led to 100% inclusion of the mutually exclusive alternate α1 exon (Figs. 1A and EV1B). The deletion of the α2 exon did not affect the splicing of alternate β exon in Mef2d transcripts in Mef2dα2 Eko or WT muscles (Fig. 1A, bottom panel). We also examined the alternative splicing of *Mef2a* and *Mef2c* transcripts and found no changes between WT and Mef2dα2 Eko muscles (Fig. EV1C). The MEF2D protein levels were similar in Mef2dα2 Eko muscles compared to WT, as assessed by immunoblotting of tibialis anterior (TA) and soleus muscle extracts in both lines (Figs. 1B and EV1D). RT-qPCR analysis in Mef2dα2 Eko mouse line 1 quadriceps and soleus revealed no changes in the mRNA levels of *Mef2d*, *Mef2a*, or *Mef2c* between the WT and Mef2dα2 Eko mice (Fig. 1C). In Mef2dα2 Eko mouse line 2, there was no change in expression of *Mef2d*, *Mef2a*, or *Mef2c* levels in quadriceps. However, in the soleus muscle, a moderate but significant increase in *Mef2c* transcripts was observed in Mef2dα2 Eko mice, in comparison to WT mice, but there was no change in the expression of *Mef2a* or *Mef2d* transcript levels (Fig. EV1E). In summary, except for the intentional deletion of the Mef2dα2 exon, we do not observe any effect on alternative splicing of other MEF2 paralogs.

## No observable differences between sedentary WT and Mef2dα2 Eko mice

The Mef2dα2 Eko mice were indistinguishable from their WT littermates, and there was no difference in the body weights of gender-matched WT and Mef2dα2 Eko mice at 9–10-weeks-age (Figs. 2A and EV2B). The tibia length normalized weights of quadriceps, soleus, TA, extensor digitorum longus (EDL), and gastrocnemius were also similar between WT and Mef2dα2 Eko mice in both lines (Figs. 2B and EV2A,B). Given the role of skeletal muscle in whole-body glucose metabolism, we performed a glucose tolerance test (GTT) in sedentary WT and Mef2dα2 Eko mice. We found no differences between the genotypes in either line (Figs. 2C and EV2C). Given the role of Mef2d in fiber-type composition (Potthoff et al, 2007b), we performed fiber-typing of EDL and soleus muscles by immunostaining for myosin heavy chain (MHC) isoforms. Our analysis revealed no significant alteration in the fiber-type composition of the muscle groups examined between the WT and Mef2dα2 Eko mice in both lines (Figs. 2D,E and EV2D,E). The number and cross-sectional area distribution of the myofibers in the EDL and soleus muscles were also similar between the WT and Mef2dα2 Eko in both lines (Figs. 2F and EV2F). Overall, there was no difference between WT and Mef2dα2 Eko sedentary mice.

## Mef2dα2 Eko mice exhibit impaired endurance running capacity

To assess differences in muscle performance, we measured all-limb grip strength normalized to body weight in Mef2dα2 Eko and age-

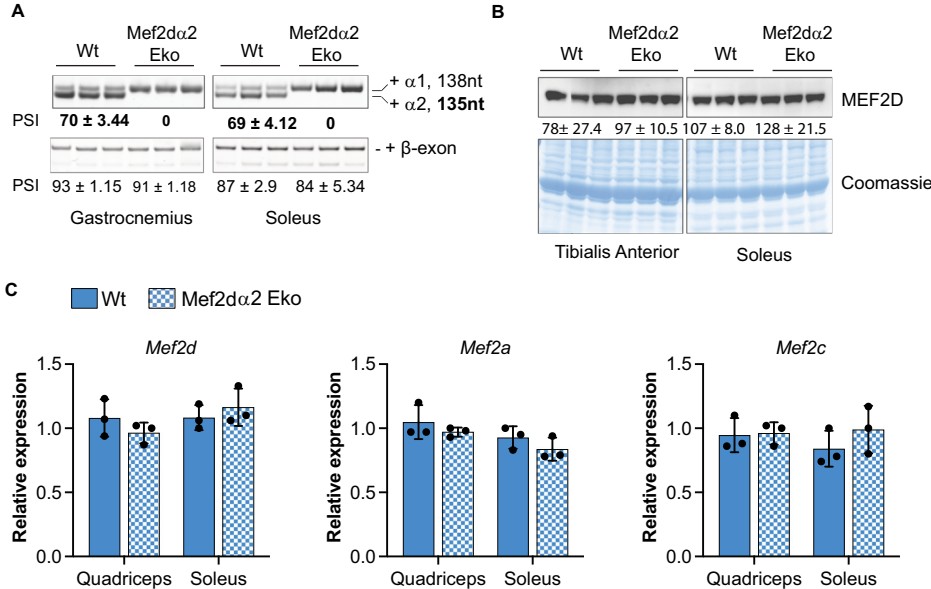

**Figure 1. Deletion of Mef2dα2 exon does not affect the expression of MEF2D protein level and expression or splicing of other MEF2 genes.**

(A) RT-PCR analysis of the mutually exclusive α-exons and alternative β exon in *Mef2d* transcript using total RNA from the indicated muscle groups from Wt and Mef2dα2 Eko mice from line 1. The numbers indicate the percent spliced in (PSI) for indicated exon. Data are mean ± SD; n = 3. Bold numbers indicate significance by Student *t* test. (B) Western blot showing MEF2D protein levels in TA (upper left) and Soleus muscles (upper right) from line 1. Bottom panels show Coomassie stained blots from the upper panels showing total protein loaded. The numbers indicate the relative protein level normalized to total protein by Coomassie stained blots. Data are mean ± SD; n = 3. (C) RT-qPCR showing mRNA levels *Mef2d*, *Mef2a*, and *Mef2c* relative to *Rpl30* using total RNA in the indicated muscle groups from line 1 mice. Data are mean ± SEM; n = 3. No significant changes were found between the genotypes by multiple *t* test in (B, C). Source data are available online for this figure.

and gender-matched WT mice. We found no differences between the genotypes in either sex in both lines (Figs. 3A and EV3A). To evaluate the running capacity in WT and Mef2dα2 Eko mice, we employed a treadmill running protocol (Fig. 3B). Following a three-day acclimatization period, WT and Mef2dα2 Eko mice were forced to run on a motorized treadmill at moderate but constant speed till exhaustion. We found that the running distance and time were ~30% reduced in Mef2dα2 Eko mice compared to WT in both lines in male animals (Figs. 3C and EV3D). However, female mice showed variable results between the genotypes in the two lines. In line 1, there was no difference in running time and distance between the genotypes, whereas, in line 2, Mef2dα2 Eko mice showed a ~20% reduction in running time before exhaustion (Fig. EV3D). In conclusion, our results show that the muscle-specific MEF2Dα2 protein isoform promotes endurance capacity in mice in a gender-biased manner.

## Glycogen storage and utilization are not affected in Mef2dα2 Eko muscles

Exercise endurance capacity is determined by the efficient import, storage (glycogen and intramuscular triglycerides), and utilization of carbohydrates, lipids, and ketone bodies in skeletal muscle (Egan and Zierath, 2013; Evans et al, 2017; Evans et al, 2022; Hargreaves and Spriet, 2018). Compared to untrained age-matched WT mice, untrained Mef2dα2 Eko male animals in both lines showed a similar decrease in running capacity (Figs. 3C and EV3D). Thus, we utilized only male animals from one line to correlate the storage and utilization of carbohydrates with reduced endurance capacity in Mef2dα2 Eko mice. We measured plasma glucose and lactate, a

byproduct of glycolysis in skeletal muscle in response to muscle activity, in sedentary and two different exercise conditions: a moderate but constant speed, similar to our endurance test (Fig. 3B), and a high-intensity increasing speed regimen. In both regimens, mice were run for a fixed time (see "Methods" section), before exhaustion, for blood glucose, lactate, and muscle glycogen levels measurements. Compared to age- and genotype-matched sedentary mice, there was a moderate, although not significant, increase in blood lactate after high-intensity exercise regimen in both WT and Mef2dα2 Eko mice (Fig. EV3C). Additionally, there was a significant increase in blood glucose in both WT and Mef2dα2 Eko mice, likely released from liver glycogen breakdown (Fig. EV3C). However, there was no significant difference in blood lactate or glucose between the WT and Mef2dα2 Eko mice in sedentary or run conditions (Fig. EV3C). When mice were run at constant but moderate speed till near exhaustion, compared to age- and genotype-matched sedentary mice, there was a significant decrease in blood glucose in both WT and Mef2dα2 Eko mice, but there was no difference between the genotypes (Fig. 3D). In addition, there was no difference in blood lactate level in either WT or Mef2dα2 Eko mice compared to sedentary mice (Fig. EV3B). We conclude that the loss of MEF2Dα2 isoform does not affect the blood glucose or lactate level in response to two different exercise regimens.

We also measured muscle glycogen levels in sedentary and exercised mice in the high-speed regimen when glycogen is a preferred substrate. Compared to sedentary mice, there was no difference in basal glycogen level between WT and Mef2dα2 Eko mice (Fig. 3E). After exercise, muscle glycogen level decreased, compared to sedentary mice, in both WT and Mef2dα2 Eko mice

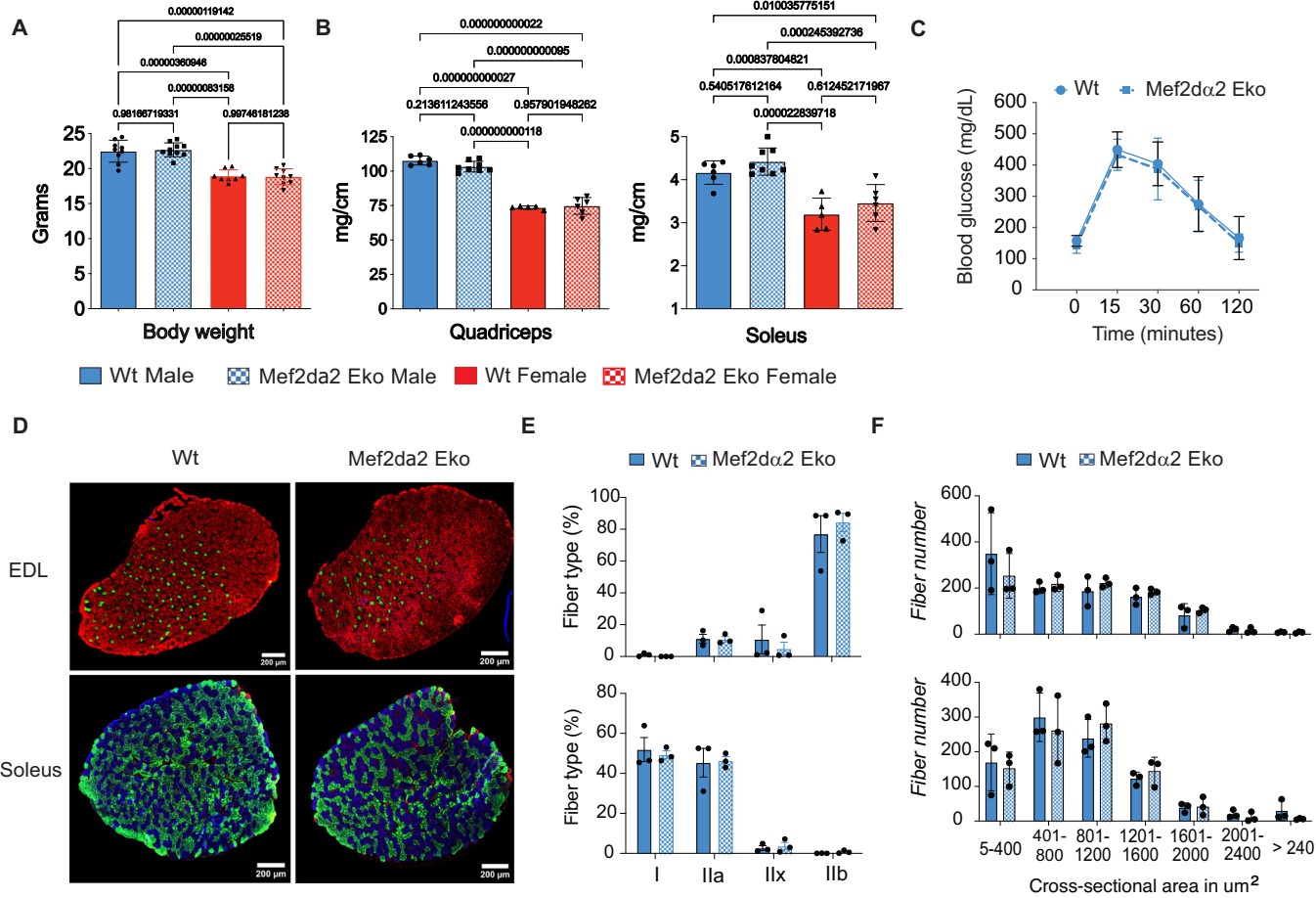

**Figure 2. Body or muscle weight, fiber type, cross-sectional area, and glucose tolerance are the same in WT and Mef2dα2 Eko mice.**

(A, B) Body and indicated tibia length normalized muscle weights from 9 to 10 weeks-old mice from line 1. Data are mean ± SD; $n \geq 8$ for body weight and $n \geq 5$ for muscle weights. n.s. not significant; $P > 0.05$, ****$P < 0.0001$, ***$P < 0.001$, *$P < 0.05$ by one-way ANOVA. (C) Glucose tolerance test or GTT 15-week-old male mice from line 1. Data are mean ± SD, $n = 9$. (D) Representative images showing cross-sections of EDL (top) and Soleus (bottom) muscles from line 1 mice immunostained for MHC-isoforms; MYH7 (Type I, blue), MYH2 (Type IIa, green), MYH4 (Type IIb, red), and MYH1 (Type IId/x, unstained). (E) Quantification of fiber-type proportion for EDL (top) and Soleus (bottom). Data are mean ± SEM; $n = 3$. (F) Distribution of cross-sectional area of myofibers in EDL (top) and Soleus (bottom) muscles from line 1 mice. Data are mean ± SEM; $n = 3$. Multiple $t$ tests were performed, and no significant genotype differences were found in (C–F). Source data are available online for this figure.

muscle (Fig. 3E). However, the difference was not statistically significant in WT mice. Nevertheless, there was no difference between the genotypes, suggesting that basal glycogen storage and its utilization during exercise is not different in WT and Mef2dα2 Eko mice muscles.

## Modest changes in glucose and fatty acid metabolic factors in Mef2dα2 Eko muscles

MEF2 transcription factors have previously been shown to regulate the expression of the gene encoding GLUT4, the predominant glucose transporter in skeletal muscle (Knight et al, 2003; Thai et al, 1998). We assessed the expression of transcripts encoding GLUT4, 10 glycolytic enzymes, lactate dehydrogenase a (LDHA) and b (LDHB) by RT-qPCR (Fig. EV3E,F). Compared to sedentary and age-matched WT muscles, glucokinase (Gck) transcripts were significantly decreased, whereas glucose-6-phosphate isomerase 1 (Gpi1) transcripts were significantly increased in sedentary Mef2dα2 Eko

muscles, but there was no change in these transcript level in response to exercise in either genotype (Fig. EV3E). Ldhb transcript levels were not different between the genotypes in sedentary or exercised conditions, but transcript levels decreased in both genotypes after exercise, albeit the decrease was statistically significant only in WT muscles (Fig. EV3F). Compared to WT, aged-matched Mef2dα2 Eko mice muscles showed a trend for higher GLUT4 protein levels by immunoblotting, but the increase was not significant in sedentary or exercised mice muscles (Fig. 3F). Additionally, the muscle *Glut4* transcript levels were not different between genotypes in sedentary or exercised muscles (Fig. EV3F).

At moderate-speed running, both carbohydrates and fatty acids are utilized as an energy source (Coyle, 1995; Muscella et al, 2020; Romijn et al, 1993). Thus, we assessed the expression of the predominant fatty acid transporter protein, CD36, in skeletal muscle by immunoblotting. Compared to WT mice, CD36 muscle expression was higher (~40%), in sedentary age-matched Mef2dα2 Eko mice (Fig. 3G). After exercise, CD36 levels increased in both WT and Mef2dα2 Eko muscles

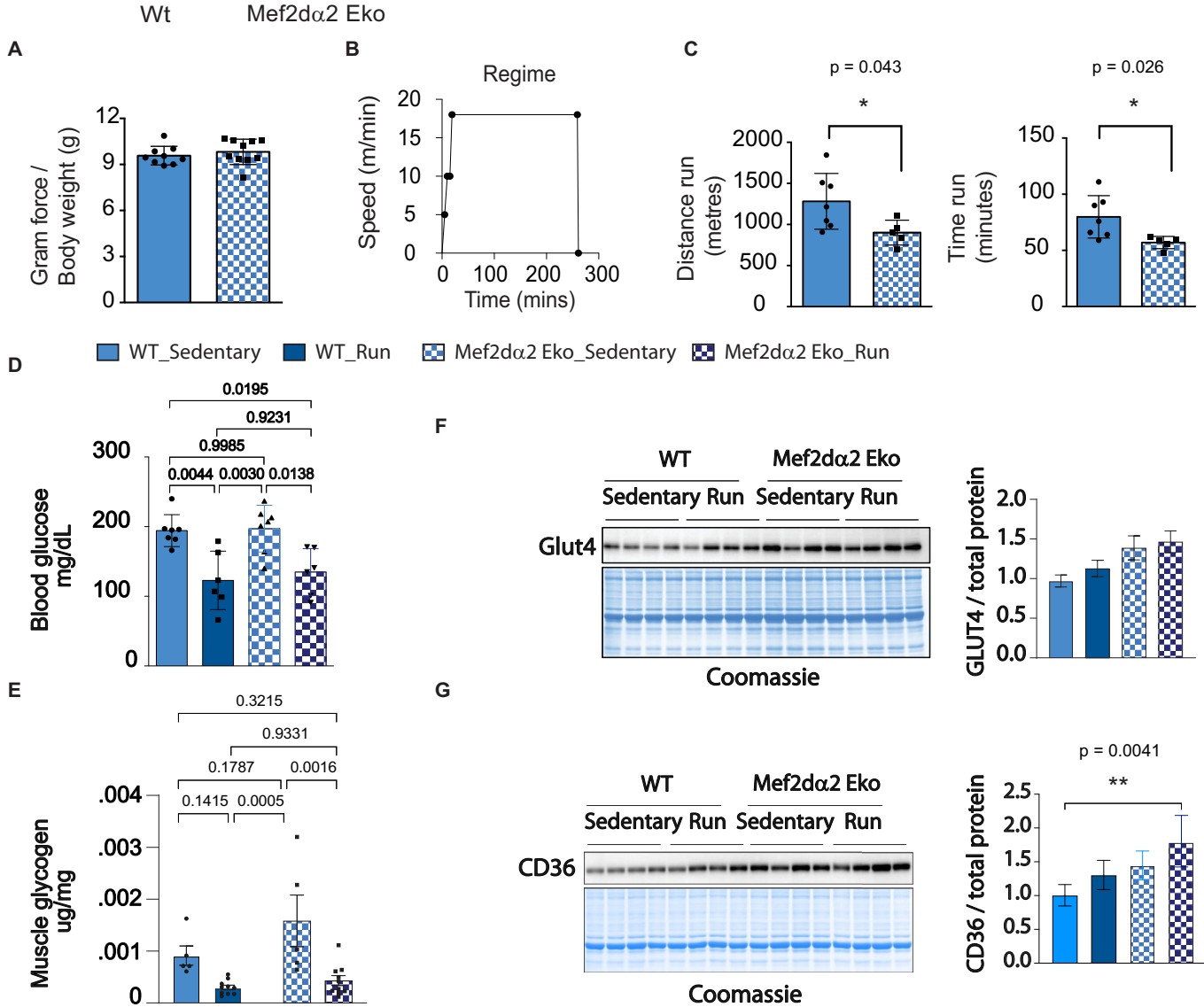

**Figure 3. Mef2dα2 Eko mice show reduced running capacity without affecting glucose and fatty acid transporter expression or muscle glycogen level.**

(A) All limb grip strength normalized to body weight from 9 to 10-weeks age male mice from line 1. Data are mean ± SD; $n \geq 9$. Student $t$ test found no differences between the genotypes. (B) A graph showing endurance treadmill running regimen used in our study. (C) Endurance capacity of indicated male mice from line 1 measured as distance run (left) and time (right) to exhaustion. Data are mean ± SD; $n \geq 5$ mice per group. *$P < 0.05$, (unpaired Student's $t$ test). The exact $P$ value is indicated above the bar graphs. (D) Blood glucose in indicated male sedentary mice and mice that were run till near exhaustion. Data are mean ± SEM; $n \geq 6$, n.s, not significant; $P > 0.05$, *$P < 0.05$, **$P < 0.01$ (one-way ANOVA). (E) Muscle glycogen level normalized to tissue weight in indicated sedentary and exercised mice. Data are mean ± SD; $n \geq 5$ mice per group. n.s, not significant; $P > 0.05$, **$P < 0.01$, ***$P < 0.001$ (one-way ANOVA). (F, G) Western blot showing GLUT4 (F) and CD36 (G) protein levels using total protein extract in indicated male sedentary mice and mice that were run till near exhaustion. Panels on the right show quantification of GLUT4 (F) and CD36 (G) proteins shown in the images on the left normalized to total protein loaded as estimated by Coomassie staining of the same blot. Data are mean ± SD; $n = 4$. **$P < 0.01$ (one-way ANOVA). The exact $P$ value is indicated above the bar graphs. Source data are available online for this figure.

(~25–30%), but there was no genotype difference (Fig. 3G). Notably, there was no difference in CD36 transcript levels between the genotypes in sedentary or exercised muscles (Fig. EV3F). We also quantified the expression of transcripts encoding 28 other fatty acid metabolism proteins involved in fatty acid transport, beta-oxidation, and regulatory function by RT-qPCR (Fig. EV3F–H). Compared to sedentary and age-matched WT muscles, Mef2dα2 Eko muscles showed significantly higher expression of transcripts encoding fatty acid binding protein 3

(FABP3) (Fig. EV3F). Compared to sedentary muscles, exercised muscles showed a moderate reduction in Ppara and Ppard transcript levels in WT mice but not in Mef2dα2 Eko mice (Fig. EV3G), whereas transcripts encoding fatty acid synthase (FASN) were upregulated after exercise in WT muscles but not in Mef2dα2 Eko muscles (Fig. EV3H). Overall, of the 24 fatty acid metabolism related transcripts, there were modest to no difference in expression between the genotypes (Fig. EV3F,H).

## Identification of transcriptome changes in Mef2dα2 Eko mice muscles

To identify the transcriptome-wide effect of Mef2d α2-exon deletion on muscle gene expression, we performed deep RNA-sequencing using polyA-selected quadriceps RNA from two sedentary littermate WT and Mef2dα2 Eko mice. On average, our RNA-sequencing yielded 139 million reads/sample of which ~80% of reads matched uniquely to the mouse genome (Table EV1). Our RNA-seq analyses also validated the targeted deletion of MEF2D α2 exon (Fig. 4A). In addition, there were 114 differentially expressed genes between WT and Mef2dα2 Eko muscles (False discovery rate or FDR or $P$ adjusted value < 0.05). Of these, 74 genes were downregulated and 40 upregulated in Mef2dα2 Eko muscles in comparison to age-matched WT muscles (Fig. 4B). We performed gene ontology analyses for genes that were downregulated using Database for Annotation, Visualization and Integrated Discovery (David) (Huang da et al, 2009), and identified enrichment of genes involved in regulating kinase, catalytic, and oxidoreductase activity, and cytoskeletal binding proteins (Fig. EV4A). Among the 40 upregulated genes, there was an enrichment of genes involved in organ development, morphogenesis, cell-cell adhesion, and wound healing response (Fig. EV4B).

Given that efficient substrate utilization for energy production is a key determinant of exercise capacity, we focused our analyses to identify genes involved in carbohydrate and fatty acid oxidation. We found that Gck transcripts were also downregulated in our RNA-seq analyses in Mef2dα2 Eko muscles (Fig. EV3E; Table EV2). However, Gck is expressed at a very low level in skeletal muscle, wherein hexokinase II is the predominant kinase for glucose phosphorylation (Printz et al, 1997; Ritov and Kelley, 2001).

## Reduced muscle expression of ketolytic genes in Mef2dα2 Eko mice causes increased systemic BHB availability

We also found *Bdh1* transcripts to be downregulated in our RNA-seq analyses in Mef2dα2 Eko muscles. Ketone bodies, primarily acetoacetate (AcAc) and 3-Hydroxybutyrate (BHB), are released into the bloodstream by the liver in response to starvation and energetic stress, including prolonged exercise (Evans et al, 2017; Evans et al, 2022). BDH1 catalyzes the interconversion of AcAc to BHB and regulates the oxidation of BHB in skeletal muscle to support muscular activity (Cox and Clarke, 2014). Our RT-qPCR analyses showed ~30% and 58% downregulation of *Bdh1* transcripts in Mef2dα2 Eko mice in soleus and TA muscles in comparison to age-matched WT mice, respectively (Figs. 4C and EV4C). Additionally, we found that *Bdh1* expression is ~14-fold higher in Soleus than TA muscle (Fig. EV4D). Thus, we performed immunoblotting using a BDH1-specific antibody and total protein from soleus muscles from WT and age-matched Mef2dα2 Eko mice, which confirmed ~35% reduction in BDH1 protein level in Mef2dα2 Eko muscles in both lines (Figs. 4D and EV4E). Furthermore, our focused analyses of two remaining muscle ketolytic enzymes found reduced OXCT1 and ACAT1 expression in the soleus muscles of Mef2dα2 Eko mice compared to age-matched WT mice (Figs. 4D and EV4E). In conclusion, these results indicate a reduced expression of all three ketolytic enzymes in the skeletal muscles of Mef2dα2 Eko mice.

It has been suggested that skeletal muscle is a major organ for ketone body oxidation at rest and during exhaustive exercise and recovery period (Balasse and Fery, 1989; Evans et al, 2017; Laffel, 1999). Thus, we hypothesized that a reduced expression of ketolytic enzymes would cause decreased BHB utilization in skeletal muscle, leading to increased systemic availability of ketone bodies. We tested our hypothesis in physiological experiments in a series of experiments using age-matched WT and Mef2dα2 Eko male mice. First, in our BHB tolerance test, Mef2dα2 Eko mice displayed reduced clearance of BHB compared to age-matched WT mice (Fig. 5A). Second, we measured BHB level immediately after forced treadmill exercise and during the recovery period, up to 3 h, in age-matched WT and MEF2Dα2 Eko mice. Compared to WT mice, the blood BHB levels were increased in MEF2Dα2 Eko mice immediately after exercise, which increased further during the recovery period, 2- and 3-h post-exercise (Fig. 5B). Lastly, we placed mice on matched control and ketogenic diet for 2 weeks. We found increased BHB levels in MEF2Dα2 Eko mice compared to age-matched WT mice (Fig. 5C). Therefore, we conclude that the MEF2Dα2 protein isoform promotes the expression of ketolytic genes in skeletal muscle during the postnatal period and adulthood, likely enhancing muscular activity through ketone body utilization.

## Reduced utilization of ketone bodies in mitochondria isolated from Mef2dα2 Eko mice muscles

Increased blood ketone body levels were observed in Mef2dα2 Eko mice in a tolerance test and after exercise, suggesting a reduced ketone body utilization within the skeletal muscle. The increased blood ketone bodies could also result from increased ketone body production by the liver. However, we didn't find any significant changes in the expression of liver ketogenic gene transcripts, including *Acat1*, *Hmgcs2*, *Hmgcl*, and *Bdh1* (Fig. EV5A). OXCT1 (also known as succinyl-CoA:3-oxoacid-CoA transferase or SCOT) is the rate-limiting enzyme for ketolysis, which uses AcAc and succinyl CoA as a substrate to produce acetoacetyl-CoA and succinic acid. Acetoacetyl-CoA is then converted to two molecules of Acetyl-CoA to feed the TCA cycle for NADH production. A study recently reported minimal utilization of ketone bodies by isolated mitochondria from skeletal muscle using Oroboros (Petrick et al, 2020). Thus, to stimulate mitochondrial utilization of AcAc, we added alpha-ketoglutarate (AKG), which generates succinyl CoA via the TCA cycle. A higher succinyl CoA level is expected to increase AcAc utilization by OCXT1, as both succinyl CoA and AcAc are needed for its activity. Using this strategy, we showed increased AcAc utilization and respiration (JO₂) using mitochondria isolated from skeletal muscle in the presence of AKG. We also found that isolated mitochondria from Mef2dα2 Eko mice displayed reduced utilization of ketone bodies and JO₂ compared to mitochondria isolated from age-matched WT mice (Fig. 5D). We also observed reduced JO₂ using mitochondria from Mef2dα2 Eko mice when AKG alone was supplied as a substrate; however, the increase in JO₂ was significant when AcAc was supplied with AKG in WT mice but not in Mef2dα2 Eko mice (Figs. 5D and EV5B). Thus, we conclude that increased blood ketone bodies in Mef2dα2 Eko mice in tolerance test and after exercise are due to reduced utilization of ketone bodies in skeletal muscle, likely contributing to reduced running capacity in Mef2dα2 Eko mice.

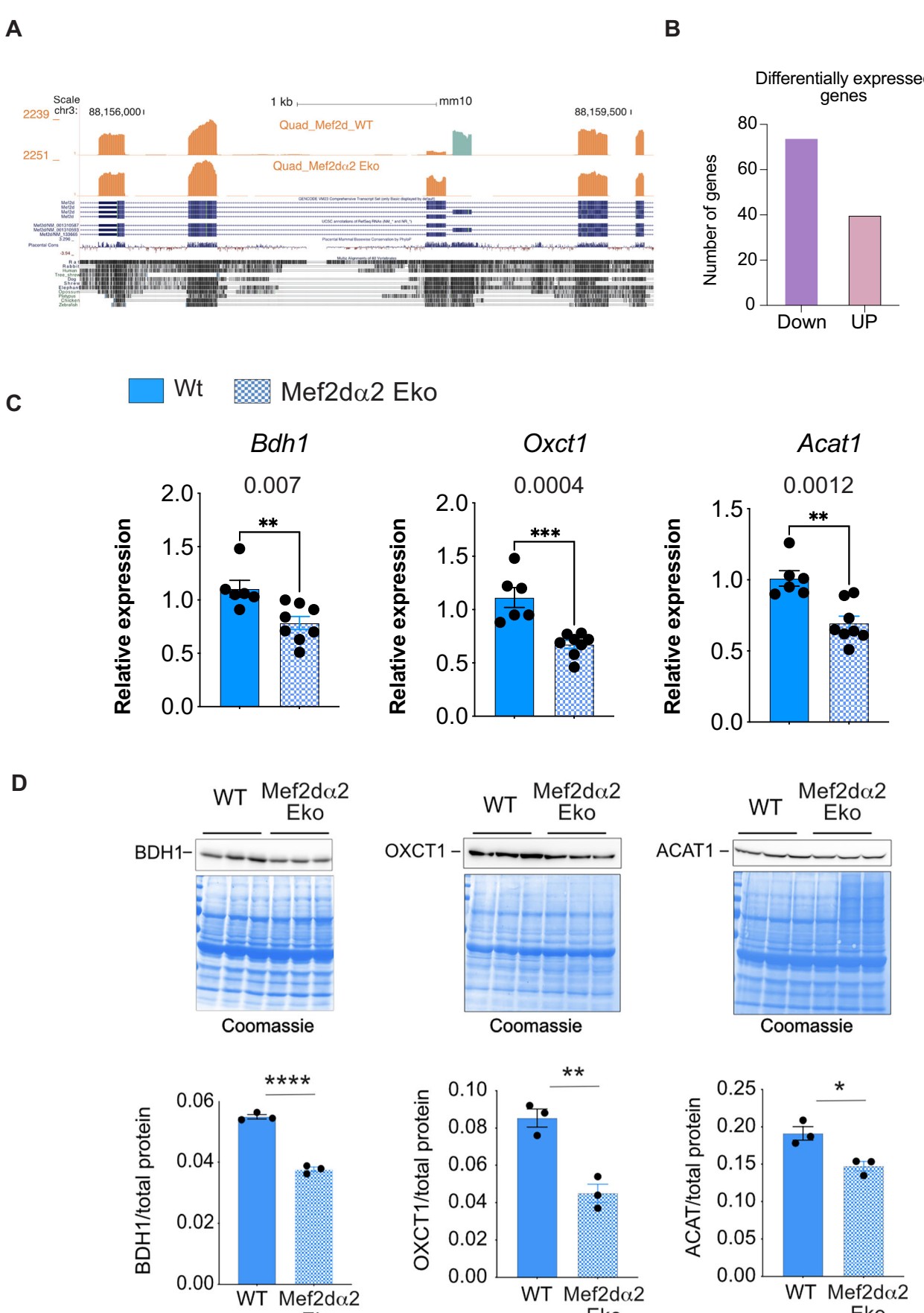

◄  **Figure 4.  Mef2dα2 Eko mice show reduced muscle expression of ketolytic genes.**

(A) RNA-sequencing data using poly-A selected RNA from WT and MEF2Dα2 Eko quadriceps muscles were aligned to mm10 UCSC browser showing *Mef2d* α exons region. (B) Graph showing number of genes that were down or upregulated in MEF2Dα2 Eko muscles in comparison to age and sex-matched WT muscles (FDR < 0.05). (C) RT-qPCR showing relative expression of *Bdh1, Oxct1, and Acat1* transcripts normalized to *Hprt* transcript levels in WT and MEF2Dα2 Eko soleus muscles from line 1. Data are mean ± SEM; $n \geq 6$. **$P < 0.01$, ***$P < 0.001$ (Student's *t* test). The exact *P* value is indicated above the bar graphs. (D) Western blot showing BDH1 level in soleus muscles of indicated mice from line 1. The panel on the right show BDH1, OXCT1, and ACAT1 level when normalized to total protein loaded as estimated by Coomassie staining of the same blot. Data are mean ± SEM, $n = 3$, *$P < 0.016$, **$P < 0.0042$, ****$P = 0.000082$ (Student's *t* test). Source data are available online for this figure.

## MEF2D binds to ketolytic genes in skeletal muscle

To determine if MEF2D can bind to the genomic regions of the ketogenic genes we analyzed publicly available ChIP-seq data from differentiated C2C12 cultures and found MEF2D occupancy within the promoter (~10 kb upstream of the gene), and gene body of Bdh1, Oxct1, and Acta1 genes (Fig. EV5C) (Gonczi et al, 2023). To determine whether MEF2D binds to ketolytic genes in mouse skeletal muscle, we performed chromatin immunoprecipitation (ChIP) in wild-type mouse muscles using an MEF2D-specific antibody. Published data from other groups have shown that the Igfn1 gene is a direct target of MEF2D, and overexpression of the MEF2α2 isoform, but not the MEF2α1 isoform, leads to a substantial increase in Igfn1 expression in differentiated C2C12 cultures. (Gonczi et al, 2023; Sebastian et al, 2013). Our RNA-seq data shows reduced Igfn1 transcript expression in Mef2dα2 Eko muscles compared to WT muscles (Table EV2). Therefore, we evaluated our MEF2D ChIP for its enrichment at the Igfn1 gene locus by qPCR analysis and found ~ a 5.6-fold enrichment of MEF2D at this Igfn1 gene body compared to the IgG control (Fig. 5E). We also observed MEF2D enriched binding in the genomic regions of Bdh1 (~4.7-fold), Oxct1 (~4-fold), and Acat1 (~9-fold), compared to IgG control (Fig. 5E). These bindings are consistent with reduced skeletal muscle expression of ketolytic genes in Mef2dα2 Eko mice. Overall, our results suggest that the MEF2Dα2 isoform, the predominant skeletal muscle isoform of MEF2D, regulates the optimum expression of ketolytic genes in skeletal muscle.

## Discussion

The highly conserved muscle-specific Mef2d α2 exon is included during the late embryonic and early postnatal period in skeletal muscle (Fig. EV1A) (Brinegar et al, 2017). Thus, it is expected that MEF2Dα2 protein isoform is likely not essential for embryonic muscle development but plays a role in adult skeletal muscle. Our extensive phenotyping assays did not find any differences at the basal level in muscle mass, fiber types, cross-section area, and glucose tolerance test (Figs. 2 and EV2). We also did not find any changes in grip strength (Figs. 3A and EV3A). However, when we performed forced treadmill assays, compared to age-matched WT mice Mef2dα2 Eko mice displayed ~30% reduction in running capacity at a constant moderate speed. The reduced running capacity in Mef2dα2 Eko mice was consistent in two independent founder lines in male animals (Fig. 3C and EV3D). However, female animals showed variable results in two lines- in one line there was no difference between the age-matched WT and Mef2dα2 Eko female animals, and in the other line female mice showed ~20% reduction in running capacity (Fig. EV3D). The relative contribution to energy generation in skeletal muscle by carbohydrates, lipids, and ketone body is dynamically regulated and primarily influenced by the intensity and duration of exercise, training status, and gender (Evans et al, 2017; Lundsgaard and Kiens, 2014; Romijn et al, 1993; van Loon et al, 2001). Thus, the variability in exercise capacity in Mef2dα2 Eko female animals may reflect intrinsic differences in substrate utilization in male and female skeletal muscle.

The blood glucose and lactate levels in WT and Mef2dα2 Eko mice were not different after two different exercise regimens (Figs. 3D and EV3B,C). In addition, the predominant glucose and fatty acid transporters, GLUT4 and CD36, respectively, showed a trend for increased expression in Mef2dα2 Eko muscles compared to WT muscles, unlikely contributing to the reduced running capacity in Mef2dα2 Eko mice (Fig. 3F,G). Our RNA-sequencing of quadriceps RNA from WT and Mef2dα2 Eko adult sedentary mice found 114 differentially expressed genes. Of these, only 47 genes show more robust differences in gene expression ($\geq$ 2-fold with P-adj $\leq$0.05). Of these genes, we focused on genes that could impact substrate utilization in skeletal muscle. We found Gck and Bdh1 to be reduced in Mef2dα2 Eko muscles when compared to WT muscles. GCK is not the major hexokinase in skeletal muscle (Magnuson, 1992). Thus, we focused on Bdh1 because it encodes the enzyme to convert BHB to AcAc, which is readily oxidized to fuel muscle activity in response to reduced carbohydrate availability (Evans et al, 2017; Laffel, 1999). A close examination of genes encoding other ketolytic enzymes, Oxct1 and Acat1, also showed reduced expression in Mef2dα2 Eko muscles suggesting for MEF2Dα2 protein isoform in regulating the postnatal ketone body metabolism. Under low carbohydrate availability conditions, ketone bodies are produced by the liver and released into the bloodstream to be utilized by peripheral tissues, including skeletal muscle, brain, and heart (Evans et al, 2017; Laffel, 1999). Skeletal muscle makes up to 40% of body mass in adult humans and accounts for the highest fraction of ketone utilization at rest (Balasse and Fery, 1989). Furthermore, ingesting ketone esters enhances muscle performance (Cox et al, 2016). The treadmill regimen we utilized led to reduced muscle glycogen, blood glucose, and lactate levels; however, there was no difference between the genotypes (Fig. 3D, E). A reduced expression of all three ketolytic enzymes in Mef2dα2 Eko muscle is expected to cause decreased BHB utilization, which is consistent with a slower clearance of BHB in Mef2dα2 Eko mice (Fig. 5A). Additionally, Mef2dα2 Eko mice also displayed increased BHB immediately after exercise, during the recovery period, and after consumption of ketogenic diet (Fig. 5B,C). The Mef2dα2 exon is not included in the Mef2d transcripts in the liver (Fig. EV1A) and not expected to affect Mef2d transcriptional program. Nevertheless, we quantified the expression of liver transcripts of Mef2d and ketogenic genes and found no significant differences between WT and Mef2dα2 Eko

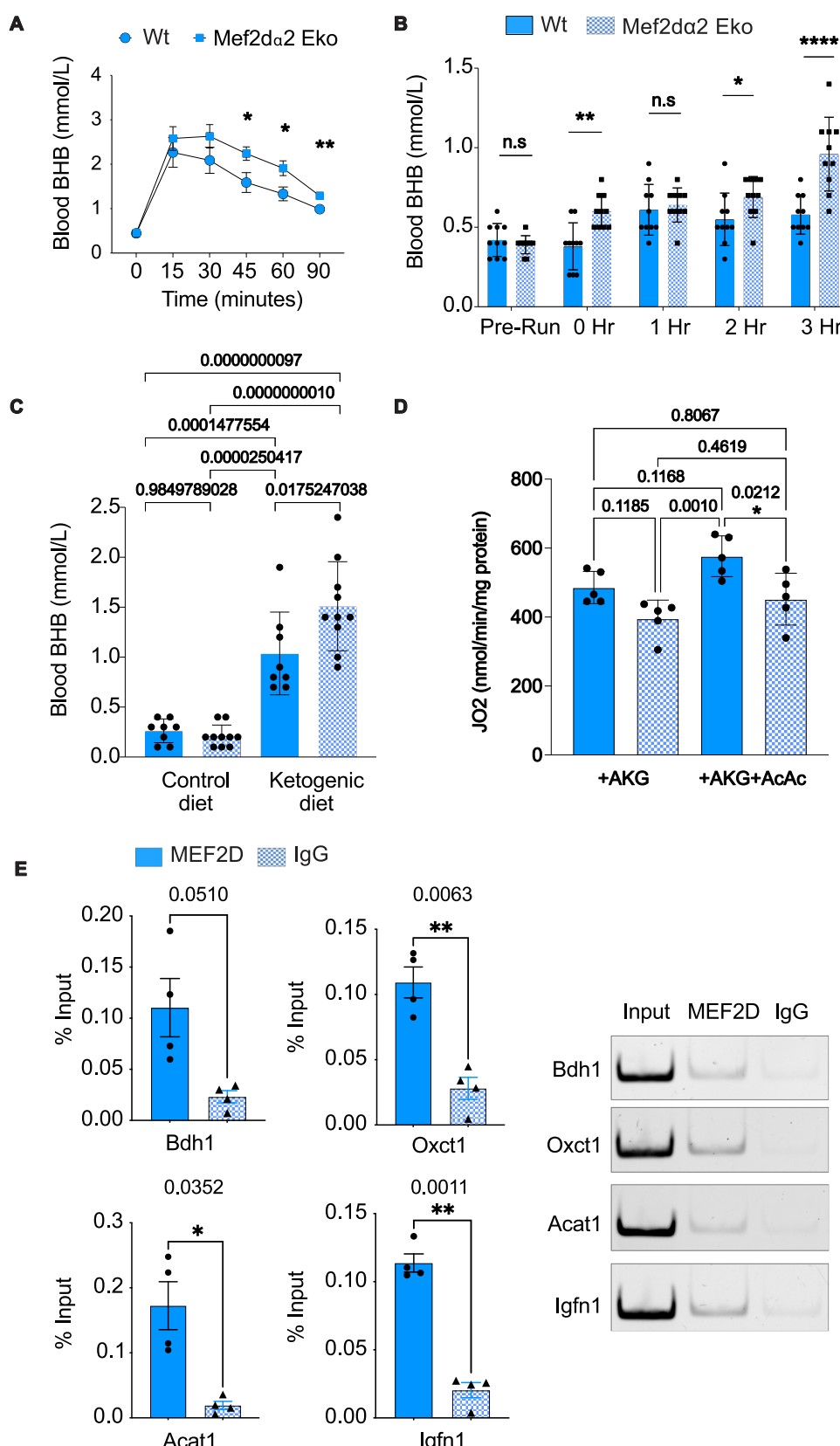

◀ **Figure 5. Mef2dα2 Eko mice show decreased muscle use of ketone bodies, and MEF2D binds to ketolytic gene regions in muscle.**

(A) β-hydroxybutyrate (BHB) tolerance test in untrained 28-week-old male sedentary mice. Graphs displays mean ± SEM, with $n \geq 9$. *$P < 0.05$ ($P = 0.02525$ at 45 min, $P = 0.021921$ at 60 min), **$P < 0.01$ ($P = 0.002777$ at 90 min) by multiple $t$ test between genotypes at different time-points. (B) The indicated age-matched male mice were run for 55 min at 18 m/min, and BHB was measured immediately after running (0 h), and 1–3 h post-run. The pre-run BHB values are from the same mice a day before the experiment. Data are mean ± SD, $n = 10$, n.s. not significant; $P > 0.05$, **$P < 0.01$ ($P = 0.00121$ at 0 h), *$P < 0.05$ ($P = 0.0485$ at 2-h post-exercise), ***$P < 0.001$ ($0.000233088$ at 3-h post-exercise) by multiple $t$ test between genotypes at different time-points. (C) The indicated age-matched male mice were fed control or ketogenic diet for 2-weeks and BHB was measured without fasting. Data are mean ± SD; $n \geq 8$. The exact $P$ value as calculated by one-way ANOVA is indicated above the bar graphs. (D) Comparison between state 3 $JO_2$ (ADP 200 μM) for isolated mitochondria from WT and MEF2Dα2 Eko skeletal muscles in presence of alpha-ketoglutarate (AKG) and AKG+Acetoacetate (AcAc). The representative time courses for these experiments are shown in Fig. EV5B. Data are mean ± SD; $n = 5$. The exact $P$ value as calculated by one-way ANOVA is indicated above the bar graphs. (E) ChIP quantitative PCR for indicated genes using MEF2D and Serotype control (IgG) are shown (left), representative DNA product visualized on a 5% acrylamide gel (right). Data are mean ± SD; $n = 4$. For Bdh1, actual $P$ value is indicated; for others, *$P < 0.05$, **$P < 0.01$ (paired Student's $t$ test). The exact $P$ value is indicated above the bar graphs. Source data are available online for this figure.

mice, suggesting that increased ketogenesis is unlikely the cause of increased blood ketone body levels in Mef2dα2 Eko mice (Fig. EV5A). Lastly, we directly measured the ketone body utilization to support respiration using isolated mitochondria from skeletal muscle using a high-resolution respirometer. Consistent with our experiments, we found reduced utilization of ketone bodies to support respiration using mitochondria isolated from Mef2dα2 Eko mice compared to age-matched WT mice. We also observed reduced respiration in Mef2dα2 Eko muscle mitochondria in comparison to WT muscle mitochondria when AKG alone was used as a substrate. Currently, we do not know the molecular mechanisms causing this phenotype. One possibility is that reduced ketone body muscle utilization may also cause a higher intramuscular level of ketone bodies, which may act as molecular signal affecting muscle metabolism and mitochondrial bioenergetics (Arima et al, 2021; Evans et al, 2017; Shimazu et al, 2013).

It was previously reported that the muscle-specific deletion of Peroxisome proliferator-activated receptor gamma coactivator 1-alpha (PGC1α) led to reduced expression of all three ketolytic genes (Svensson et al, 2016). However, PGC1α and estrogen-related receptor α (ERRα) was found to bind to Oxct1 genomic region but not in the genomic loci of Bdh1 and Acat1 genes, suggesting another transcriptional regulator plays a role in transcription of Bdh1 and Acat1 genes (Svensson et al, 2016). Our results show that MEF2Dα2 binds to all three ketolytic gene loci and likely works with other transcriptional regulators to optimize gene expression to regulate exercise capacity and adaptation, including ketolytic genes. The MEF2Dα2 exon encodes the transactivation domain of MEF2D and does affect the MADS and MEF2 domains which are important for DNA binding, dimerization, and co-factor recruitment. Previous studies using desMEF2-lacZ transgenic mice showed that this MEF2-dependent transgene is expressed throughout embryonic cardiac, skeletal, and smooth muscles but not in the adult heart and most skeletal muscles (Naya et al, 1999; Wu et al, 2000). Furthermore, the transactivating function of MEF2 proteins is induced by sustained contractile activity or motor-nerve stimulation in a calcineurin-dependent manner (Wu et al, 2000; Wu et al, 2001). However, the role of individual MEF2 paralogs or alternate MEF2 isoforms that are generated by alternative splicing in regulating exercise performance or post-exercise recovery is still not fully understood. Given the whole-body benefit exercise provides and the highly conserved role MEF2 proteins play in the development of skeletal muscle, it is crucial to understand the non-redundant role of MEF2 paralogs and protein isoforms that are generated by alternative splicing in adulthood. The blood BHB

levels increase tenfold in acute cases of diabetic ketoacidosis, which could be life-threatening (Laffel, 1999), and it has been proposed that increased skeletal muscle utilization of BHB by continuous exercise training or therapeutic strategies could alleviate the frequency of acute diabetic ketoacidosis (El Midaoui et al, 2005; Svensson et al, 2016). In summary, our findings provide new insight into the efficient utilization of the ketone body during the postnatal period and exercise training, which can be manipulated in the future to improve muscle performance or alleviate diabetic ketoacidosis.

## Methods

### Reagents and tools table

| Reagent/resource | Reference or source | Identifier or catalog number |
|---|---|---|
| **Experimental models** | | |
| Mef2dα2 Eko mice (Mef2dα2$^{e-/e-}$) | Singh lab, This study | |
| **Recombinant DNA** | | |
| **Antibodies** | | |
| MEF2D (Western blot) | BD | 610774 |
| MEF2D (ChIP) | Proteintech | 14353-1-AP |
| GLUT4 | Millipore | 07-1404 |
| CD36 | R&D Systems | AF2519 |
| BDH1 | Proteintech | 15417-1-AP |
| OXCT1 | Proteintech | 12175-1-AP |
| ACAT1 | Sigma | HPA004428 |
| MHC type I, IgG2b | DSHB | BA.D5 |
| MHC type 2a, IgG1 | DSHB | SC.71 |
| MHC type 2b, IgM | DSHB | BF.F3 |
| **Oligonucleotides and other sequence-based reagents** | | |
| PCR primers | This study | Table EV3 |
| **Chemicals, enzymes and other reagents** | | |
| **Software** | | |
| Graphpad Prism (ver. 9 & 10) | | |
| **Other** | | |

## Animals

Mice used in the present study were generated in the C57BL6/J background. For generating Mef2d-EKO mice guide, RNAs flanking the Mef2d α2 exon were co-injected with Cas9 mRNA in mouse embryos. The intron between the Mef2d α1 and α2 exon is 49 bp in length which limited us with designing guide RNAs (gRNAs) to delete α2 exon without interfering with the 5' splice sites and nearby sequences of the α1 exon. Initially, we designed 2 gRNAs targeting the intronic sequences flanking the α2 exon. However, injecting these gRNAs along with Cas9 mRNA in the pronuclear stage of mouse embryos did not yield any founder mouse with the α2 exon deletion. Thus, we designed two gRNAs targeting DNA strands near the 3'splice of the α2 exon, and the two other gRNA were designed between 60 and 120 bp downstream of the α2 exon. Deletion of the α2 exon was confirmed by PCR in eight lines of which three lines were tested and confirmed for germline transmission of the deletion. Of these three lines, Line-1047 (Line1) and Line-1056 (Line 2) were selected randomly for the present study. These mice were generated using the transgenic core facility at the Baylor College of Medicine (BCM) and were transferred to the Medical College of Wisconsin, and University of Houston after completion of materials transfer agreement. All mice procedures were approved by the Institutional Animal Care and Use Committee of the respective institutions.

## Treadmill protocols

All treadmill running experiments were performed between 2 pm and 5 pm. Before every treadmill experiment, the mice were acclimatized on Exer 3/6 six treadmill (Columbus Instruments, Ohio) for three days at a speed of 10 m/min for 9 min, followed by 18 m/min for an additional 1 min. The shock intensity was set at 2 (0.59 mA) and the shock frequency was 1 Hz. For assessing endurance capacity, mice were run at 10 m/min for 10 min followed by 18 m/min till exhaustion. The exhaustion was defined as mice sitting on the shock-grid for >5 s three times or >10 s once, whichever occurred earlier. Some mice ran in short bursts for 2–3 s and then sat on the shock-grid for 2–3 s multiple times and did not run consistently. Such mice were considered as non-performers and were not included in the analysis.

Two running regimens were used to measure blood lactate, glucose, and muscle glycogen in mice, an increasing speed, high-intensity regimen, and a moderate but constant speed regimen. In both experiments, mice were stopped before exhaustion at 32 min for a high-intensity regimen and 55 min for the moderate but constant speed regimen. For high-intensity regimen, the mice were run one after another on a high-intensity regimen starting speed of 10 m/min for 5 min, and then the speed was increased by 2 m/min every 5 min until a speed of 20 m/min (32 min). After running the mice for an additional 1 min at 20 m/min, the blood glucose and lactate were immediately measured by tail-nip bleeding using Lactate Plus analyzer (Nova Biomedical, USA). In our endurance exercise experiment at a moderate but constant speed, WT mice ran for an average of 81 min, whereas Mef2dα2 Eko mice ran for an average of 62 min. For measuring blood lactate and glucose, and muscle glycogen, mice were run one after another at a constant speed of 18 m/min but stopped before exhaustion at 55 min.

Immediately after the run lactate and glucose were measured, mice were euthanized, and muscles were harvested and snap-frozen. TA muscle was harvested first for the measurement of muscle glycogen. Glycogen content was measured in TA muscle extracts using the glycogen assay kit (BioVision). We also utilized RNA from these muscles to perform quantitative RT-qPCR to assess the expression of metabolic genes.

## RNA isolation, RNA-sequencing, and RT-qPCR analysis

RNA was isolated from mouse tissues using TRIzol reagent (Invitrogen) or RNeasy fibrous tissue mini-kit (Qiagen, Germantown, MD). For RNA-sequencing, we used quadriceps RNA from two age-matched WT and Mef2dα2 Eko adult sedentary mice. The quality of RNA was assessed by 2100 bioanalyzer (Agilent), and RNA with RIN > 7.5 was utilized for RNA sequencing using polyA selected RNAs. The RNA sequencing was performed at BCM Human Genome Sequencing Center using NovaSeq 6000 platform (Illumina). The raw fastq files were first quality checked using FastQC v0.11.8 software (https://www.bioinformatics.babraham.ac.uk/projects/fastqc/). Fastq files were aligned to mm10 mouse reference genome (GRCm38.39) and per-gene counts quantified by RSEM (1.3.1) (Li and Dewey, 2011), based on the gene annotation Mus_musculus.GRCm38.89.chr.gtf. Differentially expressed genes are called with FDR < 0.05 using DESeq2(Love et al, 2014). For RT-qPCR After removing genomic DNA contamination by treatment with amplification grade DnaseI (Invitrogen), 2 μg of RNA was converted to cDNA using the High-Capacity cDNA Reverse Transcription kit (ThermoFisher). RT-PCR was performed on cDNA using primers flanking the Mef2d α exons and the products were resolved on 5% polyacrylamide gels. The gels were imaged on a ChemiDoc MP imager (Bio-Rad), and the PSI was calculated as previously described (Singh et al, 2014). Quantitative PCR (qPCR) was performed using iQ SYBR Green supermix on a CFX384 system (Bio-Rad). Data analysis was performed using CFX Manager software (Bio-Rad), and the relative expression was calculated using the comparative Ct method (Schmittgen and Livak, 2008). The list of primers and their sequences used in this study are in Table EV3.

## Preparation of total extracts and Western blotting

Total protein extracts were prepared by homogenizing frozen muscles in lysis buffer (50 mM Tris–HCL, pH 7.5, 100 mM NaCl, 10 mM EDTA, 10 mM EGTA, 10% glycerol, 1% NP40, 50 mM NaF, 10 μM MG132, 1mMPMSF, 0.5% Sodium deoxycholate, 1% SDS, protease and phosphatase inhibitor (Roche). Equivalent amounts of proteins, as estimated with BCA assay kit (Pierce), were separated by SDS-PAGE, and transferred onto PVDF membranes. After blocking with 5% Milk-TBST (0.1% Tween) for 1 h at room temperature, membranes were incubated overnight at 4 °C with primary antibodies against MEF2D (BD, 610774), GLUT4 (Millipore, 07-1404), CD36 (R&D Systems, AF2519), BDH1 (Proteintech, 15417-1-AP), OXCT1 (Proteintech, 12175-1-AP), and ACAT1 (Sigma, HPA004428). Subsequently, the membranes were washed with TBST (3 × 10 min) and incubated with secondary antibodies conjugated to HRP for 1 h at room temperature followed by 3 × 10 washes with TBST. Signals were developed using SuperSignal West Pico substrate (Pierce) and imaged on a

ChemiDoc MP imager (Bio-Rad). The images were analyzed using the imagelab software (Bio-Rad).

## Fiber typing and cross-section area calculations

Freshly isolated EDL and Soleus muscles were frozen with OCT (Tissue-Tek) in isopentane cooled in liquid nitrogen. 10 µM longitudinal sections were cut and placed on a glass slide. Dried sections were washed with PBS ($2 \times 2$ min). The slides were blocked for 1 h with 5% (v/v) normal goat serum (NGS) in TBST, and then incubated overnight at 4 °C with a mix of primary antibodies diluted with 5% NGS in TBST. The primary antibodies used were, BA.D5 (MHC type I, IgG2b), SC.71 (MHC type 2a, IgG1), and BF.F3 (MHC type 2b, IgM) from DSHB, and Anti-Dystrophin from Abcam (ab15277, to stain fiber periphery). Subsequently, the sections were washed with PBS ($3 \times 5$ min) and incubated with a mix of secondary antibodies diluted with 5% NGS in TBST for 1 h at room temperature. The secondary antibodies used were, Goat anti-Mouse IgG2b (DyLight 405) from Jackson Immunoresearch, Goat anti-Mouse IgG1 (Alexa 488), Goat anti-Mouse IgM (Alexa 555), and Goat anti-Rabbit IgG, (Alexa 633) from Thermofisher. The sections were again washed with PBS ($3 \times 5$ min) and then mounted using ProLong™ Diamond Antifade Mountant without DAPI (Thermo Fisher). The images were scanned using VS120 whole slide scanner (Olympus USA) at ×20 magnification to capture the entire muscle section. The 16 bit images were imported into Visiopharm software (Visiopharm, Denmark) for automated fiber typing that uses morphometric measurement methods to automatically measure fiber size and min ferret distance. Dystrophin stain (Alexa 633-purple color) was used as mask for fiber outlines. Fibers, type I, IIa, IIb and IIx/d (empty fibers) were identified by the secondary antibody fluorophore blue (dylight 405), green (Alexa 488), red (Alexa 555) or empty. Following classification and processing to eliminate classified regions that are non-specific, and fibers smaller than 5 µm. Each fiber type was counted (area and diameter) and the data was plotted percentage of fiber types and distribution of fiber cross section area between the genotypes ($n = 3$).

## Intraperitoneal Glucose tolerance test (IPGTT)

For IPGTT, mice were fasted for 6 h by removing food from the cages in the morning. Mice were then weighed and restrained, and baseline blood glucose ($t = 0$) was measured by the tail-nip bleeding using glucometer. The mice were then injected intraperitoneally with a 30% glucose solution (1.5 g/ kg body weight) and blood glucose was measured at $t = 15$-, 30-, 60-, and 120-min post-injection.

## BHB tolerance test and Blood BHB measurement after exercise, ketogenic diet feeding

For the BHB tolerance test, mice fasted for 5 h, and baseline blood BHB ($t = 0$) was measured by the tail-nip bleeding using STAT-Site® WB Meter (Stanbio Laboratory, Boerne, TX). The mice were then injected intraperitoneally with 1.5 g/ kg body weight of BHB solution (H6501, Millipore Sigma). Blood BHB was measured at $t = 15$-, 30-, 60-, and 90-min post-injection. To determine the BHB level after exercise, we mice ran the mice for 55 min at 18 m/min, and BHB was measured immediately after the run (0 h) and 1–3 h post-run. To determine the BHB level after ketogenic diet feeding,

we fed mice control (F1515) or Ketogenic Diet (F3666) from Bio-Serv, NJ, USA, for 14 days, and BHB was measured without fasting.

## Grip strength test

All-limb grip test was performed using a Grip strength meter (Columbus Instruments, model: 1027SM Grip Strength Meter with Single Sensor) that includes a grid connected to a force transducer and a digital display screen. The mouse to be tested was removed from the cage and placed on the grid. Held by the tail the mouse is allowed to grab the grid with all four paws and then pulled horizontally backward gently. The maximum force by which the mouse gripped the grid just before losing the grip is displayed as peak tension and is noted. Each mouse is tested three times with 10 min gap in between. During this gap, the grid is cleaned with 70% ethanol, and the remaining mice are tested one after another.

## Mitochondrial isolation and protein assay

Skeletal muscles mitochondria were isolated through differential centrifugation method, as described previously(Koves et al, 2023; Petrick et al, 2020; Tomar et al, 2022), with slight modifications. Briefly, the skeletal muscles (TA, EDL, quadricep and gastrocnemius of both legs) from both WT and Mef2dα2 Eko mice were extracted and was immediately placed in isolation buffer (IB) containing (in mM) 200 mannitol, 50 sucrose, 5 $KH_2PO_4$, 5 3-(N-morpholino) propane sulfonic acid (MOPS), and 1 EGTA, with 0.1% bovine serum albumin (BSA) at pH 7.15 (adjusted with KOH). The skeletal muscles were chopped and minced in 1 ml of ice-cold IB. Then the minced skeletal muscles were suspended in 2 ml IB with 5 U/ml protease and homogenized for 60 s. Next, 23 ml IB was added, and the suspension of tissue was homogenized further for 30 s. The homogenized suspension was centrifuged at $800 \times g$ for 5 min at 4 °C. The supernatant containing mitochondria was passed through the 70- and then 40-micron filter and was centrifuged at $9000 \times g$ for 10 min at 4 °C. Mitochondrial pellets were suspended in 1.2 ml IB using an ice-chilled Teflon pestle and centrifuged at $9000 \times g$ for 3 min at 4 °C. Isolation Buffer was aspirated, and mitochondrial pellets were suspended in 150 µl of IB. Protein concentration was determined with using Bio-Rad Quick Start Bradford Assay Kit, and the mitochondria were used at a final concentration of 0.05 mg/ml for all Oxygraph-2k experiments.

## Mitochondrial O$_2$ consumption measurement

Isolated mitochondrial respiration was studied by sequentially increasing ADP concentration protocol as described before (Tomar et al, 2022). Using an Oxygraph-2k (O2k) system (Oroboros Instruments, Innsbruck, Austria) and associated DatLab 7 software for data collection and analysis, mitochondrial O$_2$ consumption (respiration) was measured. Mitochondria from WT and Mef2dα2 Eko mice (0.05 mg protein/ml) were suspended in the O2k chamber (2 ml) having respiration buffer (RB) containing (in mM) 130 KCl, 5 $K_2HPO_4$, 20 MOPS, 1 EGTA, and 0.1% BSA at pH 7.15 adjusted with KOH. Prior to every experiment, the air in the O2k chambers was allowed to acclimate to 37 °C for 2–3 min, or until a steady signal was achieved at an O$_2$ concentration of roughly 205 µM. Isolated mitochondria from both the mice strains were suspended parallelly in RB to start studies at t = 0 min in order to reach state 1.

Inside the O2k chambers, the mitochondrial suspension was continuously stirred (750 rpm), and chemicals were administered using Hamilton syringes through the titanium injection port of the stoppers. Alpha-ketoglutarate (AKG) (5 mM) and a combination of AKG (5 mM) + acetoacetate (AcAc) (10 mM), which are saturating concentrations of substrates, were used to start the state 2 respiration. At time $t = 4$ min, 100 μM ADP was added to measure state 3 respiration. Following the attainment of state 4 respiration, an increasing concentration of ADP (200 μM) was added (e.g., when all the added ADP was converted to ATP). In terms of protein, oxygen consumption rates (OCR; $JO_2$) were reported as nmol of $O_2$/min/mg. With the DatLab 7 program, the slope of the $O_2$ concentration data was determined by averaging five data points, which were recorded every second.

### Chromatin immunoprecipitation (ChIP)

The ChIP was performed as previously described with some minor modifications as described below (Joshi et al, 2017). For each biological replicate, three mice were euthanized and hindlimb muscles—including the tibialis anterior, soleus, gastrocnemius, extensor digitorum longus (EDL), and quadriceps—were isolated and minced in hypotonic buffer (10 mM Tris-HCl, pH 7.3; 10 mM KCl; 5 mM $MgCl_2$; 0.1% NP-40, freshly added protease inhibitor (Sigma 4693159001). Minced tissue was incubated on ice for 8 min and then homogenized for 30 s using a mechanical tissue homogenizer (OMNI International). Tissue was then fixed with 1% formaldehyde (Thermo Scientific, 28908) for 10 min and quenched with 125 mM glycine (Sigma G7126) for 10 min, shaking at room temperature. The fixed lysate was further homogenized using an ice-cold loose Dounce homogenizer and centrifuged at 1000 ×$g$ for 5 min at 4 °C. The pellet was resuspended in hypotonic buffer and filtered sequentially through 70-micron and 40-micron cell strainers. The filtered lysate was centrifuged at 1000 ×$g$ for 5 min at 4 °C. The pellet was either stored at −80 °C or resuspended in SDS nuclei lysis buffer (50 mM Tris-HCl, pH 8.0; 10 mM EDTA; 1% SDS) with protease inhibitor cocktail (Aprotinin (Sigma A6279), Pepstatin A, Leupeptin, PMSF (Sigma 93482) and sonicated for 11 min (30 s on/30 s off). Sonication efficiency was checked before immunoprecipitation. After sonication, the chromatin lysate was diluted tenfold with ChIP dilution buffer (200 mM NaCl, 50 mM Tris-HCl, pH 8.0; 5 mM EDTA; 0.5% NP-40). One-tenth of the diluted lysate was saved at −80 °C as input control. The remaining lysate was incubated overnight at 4 °C on a rotator with MEF2D antibody (Proteintech 14353-1-AP) or IgG control antibody (Cell Signaling Technology 2729S). Pre-washed magnetic protein G beads (New England Biolabs S1430S) were added and incubated with the antibody-antigen complex for 2 h at 4 °C on a rotator. Beads were washed sequentially with the following buffers: Low salt wash buffer (20 mM Tris-HCl, pH 8.0; 2 mM EDTA; 150 mM NaCl; 0.1% SDS; 1% Triton X-100) with protease inhibitor cocktail, High salt wash buffer (same as above, with 500 mM NaCl) with protease inhibitor cocktail, LiCl wash buffer (10 mM Tris-HCl, pH 8.0; 1 mM EDTA; 250 mM LiCl; 1% NP-40; 1% sodium deoxycholate) with protease inhibitor cocktail, and TE buffer. This complex was treated with 1.5 μL RNase A (Thermo Scientific EN0531) at 37 °C for 30 min, followed by the addition of 7.5 μL 20% SDS and 3.25 μL Proteinase K (New England Biolabs P8107S) and incubated at 37 °C for 9.5 h. Samples were then transferred to 65 °C for 6 h to reverse cross-links. DNA was purified using a spin column kit (Cell Signaling Technology 14209S). Quantitative PCR was performed with PowerTrack SYBR Green Master Mix (Sigma) on a CFX Opus 384 Real-Time PCR system (Bio-Rad). Primers targeting Bdh1, Oxct1, Acat1, and Igfn1 regions were used.

### Statistics

The graphs or plots are presented as mean ± SD (standard deviation) or mean ± SEM (standard error of the mean) with sample size ≥3 and indicated in the figure legend. The datasets were not tested for normality. All statistics were performed using GraphPad Prism or R. For comparison of the two groups, Student $t$ test or multiple $t$ tests was performed. For the comparison of >2 groups, one-way ANOVA was used, followed by Tukey's post-hoc analysis. For comparing the relative expression of multiple genes by RT-qPCRs in two groups in sedentary and run mice muscles, we utilized two-way ANOVA, followed by Tukey's post-hoc analysis. The $P$ values or adjusted $P$ values are indicated as *$P < 0.05$, **$P < 0.01$, ***$P < 0.005$, ****$P < 0.0001$.

## Data availability

The RNA-seq data using RNA from quadriceps muscle from two wild-type and Mef2dα2 Eko mice were deposited into the Gene Expression Omnibus under accession number GSE302518.

The source data of this paper are collected in the following database record: biostudies:S-SCDT-10_1038-S44319-025-00578-3.

## Peer review information

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

## Acknowledgements

We thank Dr. Jason Jarzembowski, the chair of the pediatric pathology division, Department of Pathology (MCW), for his unwavering support for this project. The authors acknowledge Dr. Thomas Cooper's lab for help with initial mice phenotyping. The authors also thank Dr. John Lough, Dr. Brian Link, Dr. Joy Lincoln, and Dr. John Imig for discussions and manuscript feedback. This study was supported by startup funds from the Medical College of Wisconsin (MCW) and University of Houston UH), pilot funding from the Research Affairs Committee (MCW) and Children's Wisconsin Research Institute, and funding from the American Heart Association Scientist Development Award (15SDG25610021) and NIH (5R21AR083158) to RKS.

## Author contributions

**Sushil Kumar**: Data curation; Formal analysis; Methodology; Writing—original draft. **Xuan Ji**: Data curation; Formal analysis; Methodology. **Hina Iqbal**: Data curation; Formal analysis; Methodology. **Xiangnan Guan**: Resources; Data curation; Formal analysis; Investigation; Visualization; Methodology. **Brittany Mis**: Data curation; Formal analysis; Methodology. **Devanshi Dave**: Data curation; Formal analysis; Methodology. **Suresh Kumar**: Data curation; Formal analysis; Methodology. **Jacob Besler**: Data curation; Formal analysis; Methodology. **Ranjan Dash**: Supervision; Methodology; Writing—original draft. **Zheng Xia**: Conceptualization; Resources; Data curation; Formal analysis; Supervision; Investigation; Methodology; Writing—original draft. **Ravi K Singh**: Conceptualization; Resources; Data curation; Formal analysis; Supervision; Funding acquisition; Validation; Investigation; Methodology; Writing—original draft; Project administration; Writing—review and editing.

Source data underlying figure panels in this paper may have individual authorship assigned. Where available, figure panel/source data authorship is listed in the following database record: biostudies:S-SCDT-10_1038-S44319-025-00578-3.

## Disclosure and competing interests statement

The authors declare no competing interests.

# Expanded View Figures

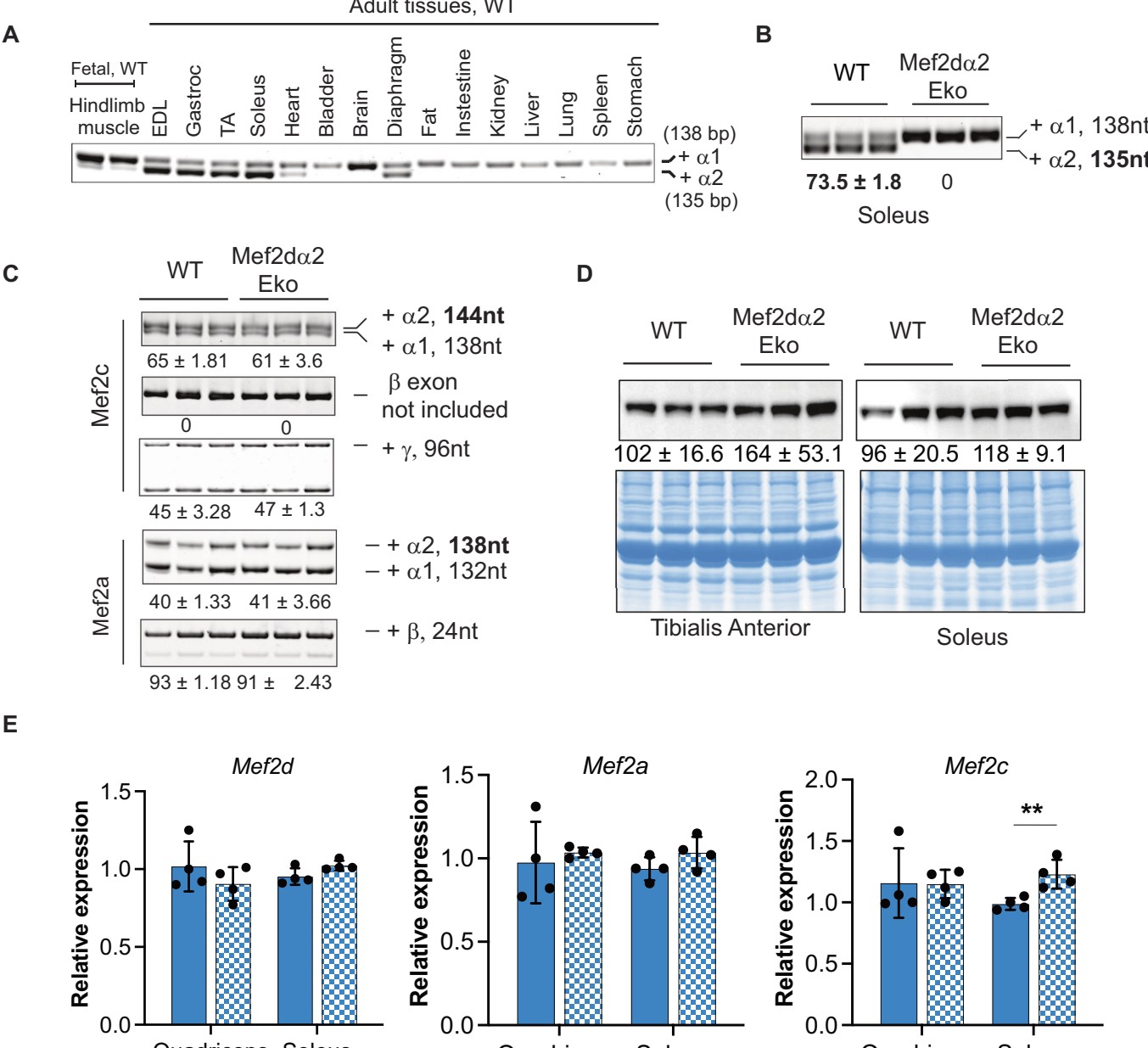

**Figure EV1. Deletion of Mef2dα2 exon does not affect the expression or splicing of other MEF2 genes.**

(A) RT-PCR analysis of the mutually exclusive α-exons in *Mef2d* transcript using total RNA from the indicated fetal hind limb muscle and tissues from adult wild-type mice. (B) RT-PCR analysis *of Mef2d* α exons using soleus RNA from line 2. The numbers indicate PSI; data are mean ± SD; $n = 3$. The numbers indicate the percent spliced in (PSI). Bolded numbers are significant by Student *T* test. (C) RT-PCR analysis of the alternative exons in *Mef2c* and *Mef2a* transcripts using total RNA from gastrocnemius muscle from line 1. Data are mean ± SD; $n = 3$. PSI for exon is indicated and was not significantly different between the genotypes by Student's *t* test. (D) Western blot showing MEF2D protein levels in TA (upper left) and Soleus muscles (upper right) from line 2. The bottom panels show Coomassie stained blots, from the upper panels showing total protein loaded. The numbers are mean ± SD; $n = 3$. Student's *t* test found no differences in genotypes. (E) RT-qPCR showing relative mRNA levels *Mef2d*, *Mef2a*, and *Mef2c* relative to *Rpl30* using total RNA in the indicated muscle groups from line 2 mice. Data are mean ± SEM; $n = 4$. **$P = 0.0088$ (multiple Student's *t* test, unpaired). Source data are available online for this figure.

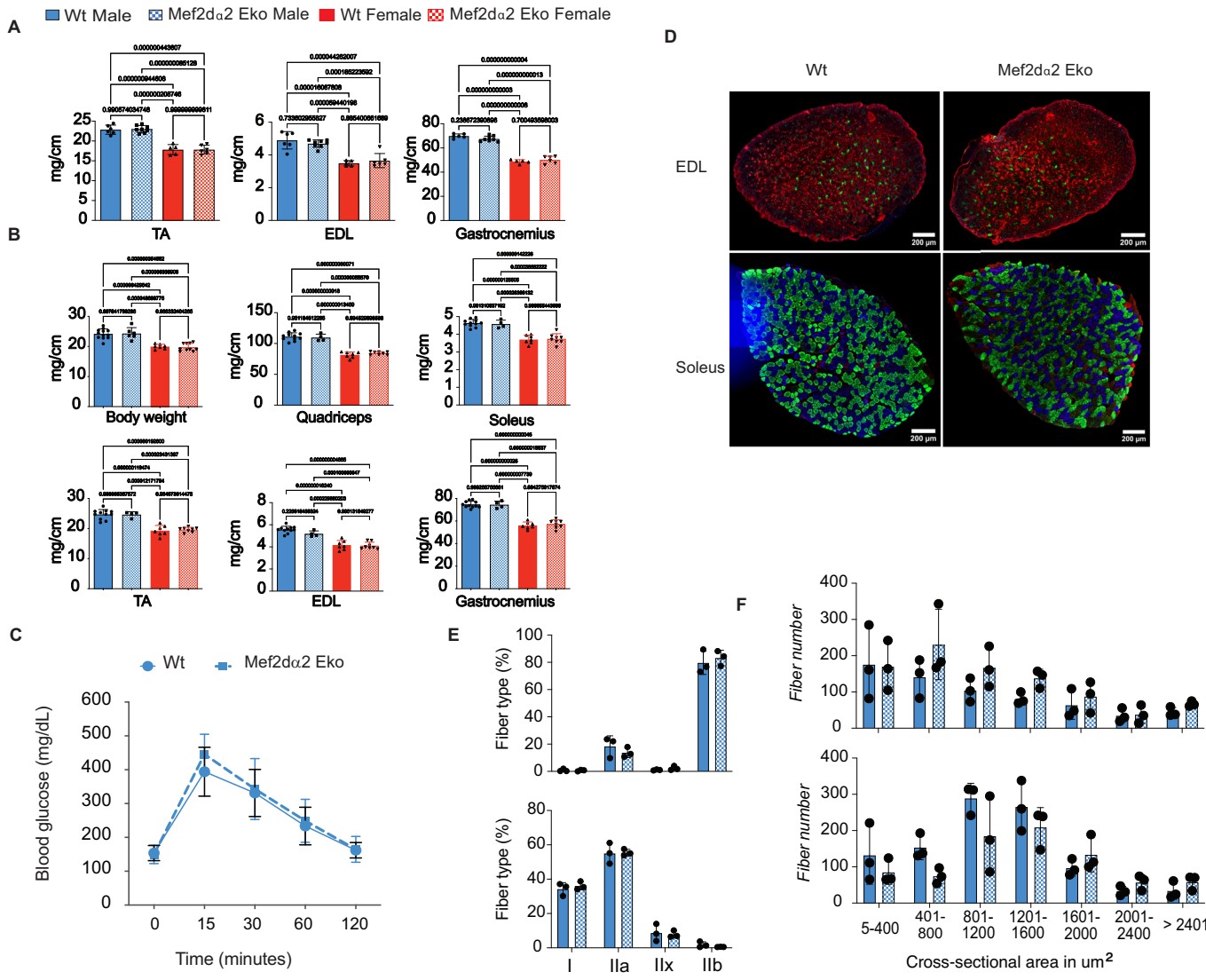

**Figure EV2.   No observable phenotypes in Mef2dα2 Eko mice.**

(A) Isolated muscle weights of the indicated muscle groups from line 1 mice normalized to tibia length. Data are mean ± SD; $n \geq 5$. ****$P < 0.0001$ by one-way ANOVA. (B) Body and indicated muscle weights of 7-month-old mice normalized to tibia length from line 2. Data are mean ± SD; $n \geq 4$. ****$P < 0.0001$, ***$P < 0.001$ by one-way ANOVA. (C) GTT in line 29-weeks-old mice from line 2. Data are mean ± SD, $n = 9$. (D) Representative images showing cross-sections of EDL (top) and Soleus (bottom) muscles from line 2 mice immunostained for MHC-isoforms; MYH7 (Type I, blue), MYH2 (Type IIa, green), MYH4 (Type IIb, red), and MYH1 (Type IId/x, unstained). (E) Quantification of fiber-type proportion in EDL (top) and soleus (Bottom). Data are mean ± SEM; $n = 3$. (F) Distribution of cross-sectional area of myofibers in EDL (top) and Soleus (bottom) muscles from line 2 mice. Data are mean ± SEM; $n = 3$. Multiple $t$ tests were performed, and no significant genotype differences were found in (C, E, F). Source data are available online for this figure.

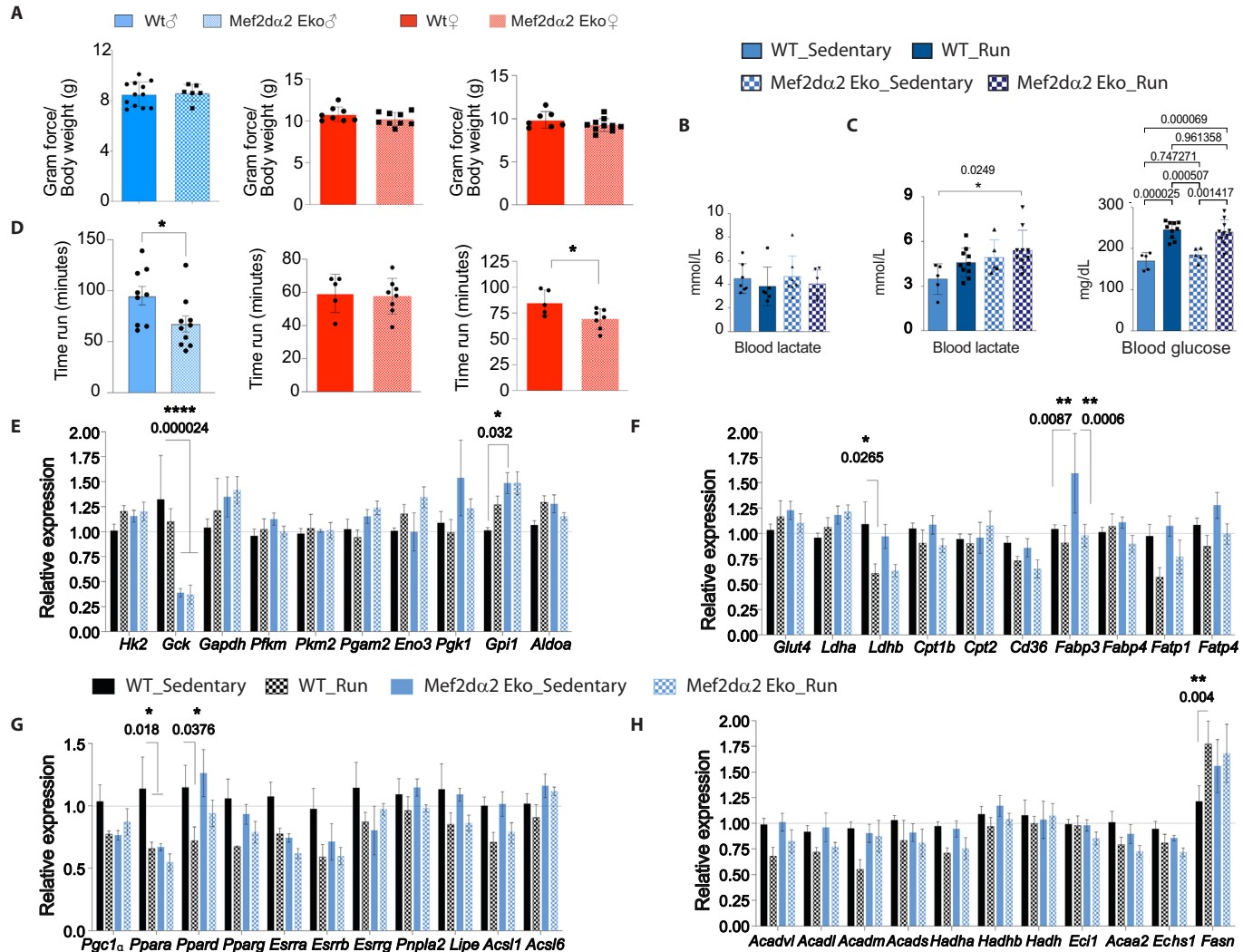

**Figure EV3. Mef2dα2 Eko mice show moderate changes in muscle expression of genes involved in glucose and fatty acid metabolism.**

(A) All limb grip strength of indicated male mice from line 2 when normalized to body weight at 9–10-weeks age (left panel, $n \geq 6$). All limb grip strength, normalized to body weight, of indicated age-matched female mice from line 1 (middle panel, $n \geq 8$) and line 2 (right panel, $n \geq 7$). Data are mean ± SD. Multiple $t$ test found no differences between the genotypes. (B) Blood lactate levels in indicated sedentary mice and mice run at constant moderate speed till near exhaustion from line 1. Data are mean ± SD; $n \geq 5$. No genotype differences found by one-way ANOVA. (C) Blood lactate (left) and glucose (right) levels in indicated mice and mice run using increasing speed or high-intensity protocol. Data are mean ± SD; $n \geq 5$. The exact $P$ value is indicated above the bar graphs. (D) Endurance capacity of line 2 male mice measured as time to exhaustion during a treadmill running protocol (left panel, $n \geq 9$), Data are mean ± SEM; $n \geq 9$ mice per group. *$P = 0.0326$ by unpaired Student's $t$ test. Time to exhaustion for indicated female in line 1 (middle panel, $n \geq 5$, no difference between the genotype by unpaired Student's $t$ test) and line 2 (right panel, $n \geq 5$, *$P = 0.0293$ by unpaired Student's $t$ test). Data are mean ± SD. Gene expression analysis in sedentary and run mice subjected to a moderate-intensity treadmill running protocol till near exhaustion. Muscles were harvested immediately after exercise. RT-qPCR showing mRNA levels of genes involved in glycolysis (E), glucose and fatty acid transport (F), and FA metabolism (G, H) relative to $Rpl30$ in indicated mice. Data are mean ± SEM; $n = 4$. (*$P < 0.05$, **$P < 0.01$, two-way ANOVA). Selected $P$ values are indicated above the bar graphs. Source data are available online for this figure.

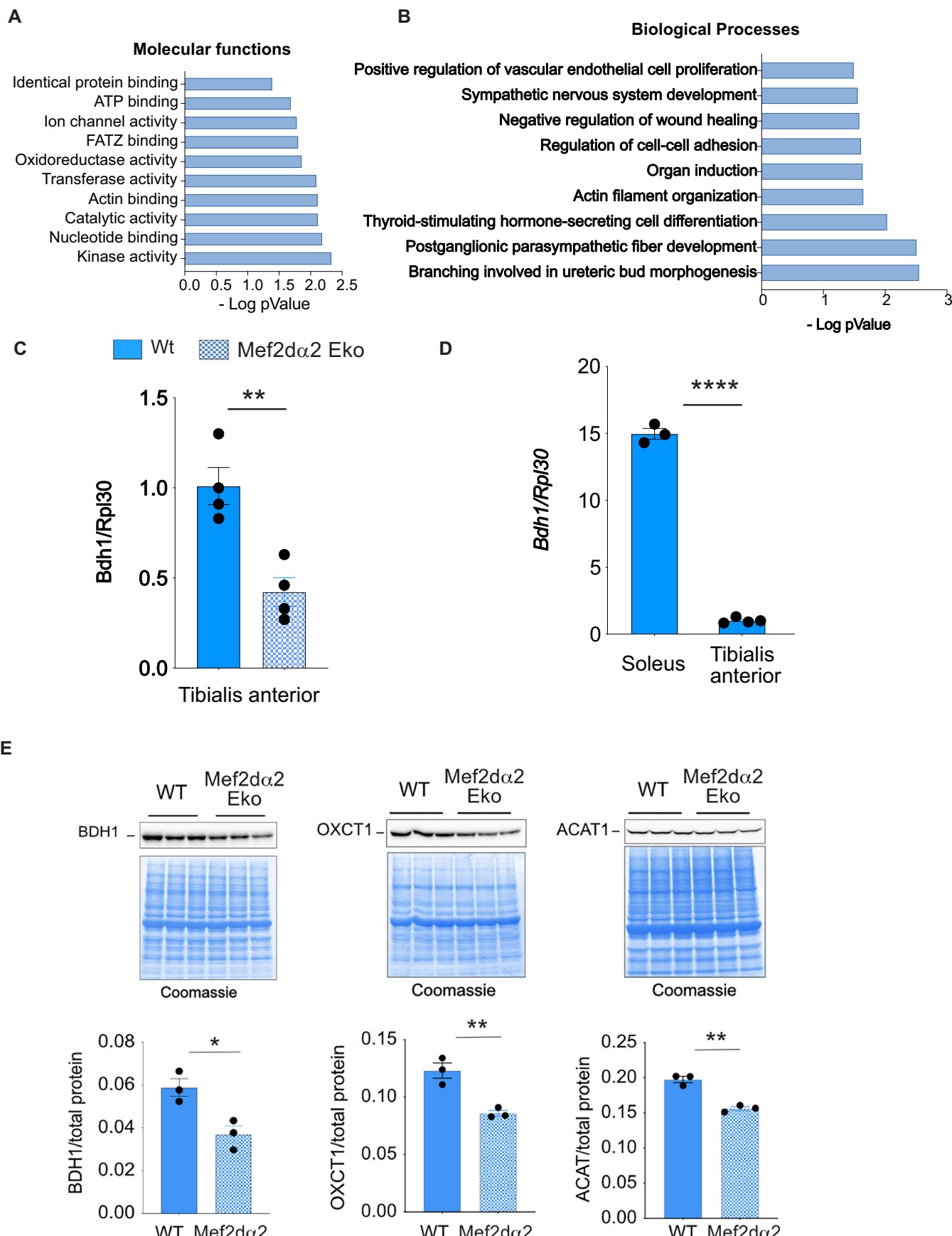

◀ **Figure EV4. Mef2dα2 Eko mice show reduced muscle expression of ketolytic enzymes.**

Gene ontology analysis using DAVID showing enrichment of molecular functions (**A**) and biological processes (**B**) among genes that are downregulated in MEF2Dα2 Eko muscles with *P* value < 0.05 using Fisher's exact test. (**C**) RT-qPCR showing relative expression of *Bdh1* normalized to *Rpl30* transcript levels in WT and MEF2Dα2 Eko TA muscles from line 1. Data are mean ± SEM; *n* = 4. \*\**P* < 0.01 (*P* = 0.004 by unpaired Student's *t* test). (**D**) Relative expression of *Bdh1* transcripts normalized to *Rpl30* in indicated muscles from WT mice. Data are mean ± SEM, *n* = 3, \*\*\*\**P* < 0.0001, (*P* = 0.00000021 by unpaired Student's *t* test). (**E**) Western blot showing BDH1, OXCT1, and ACAT1 level in soleus muscles of indicated mice from line 3. The panels below show BDH1, OXCT1, and ACAT1 level when normalized to total protein loaded as estimated by Coomassie staining of the same blot. Data are mean ± SEM, *n* = 3, \*\**P* < 0.05 (*P* = 0.0186 for BDH1), \*\**P* < 0.01 (*P* = 0.0065 for OXCT1, and *P* = 0.0013 for ACAT1 by unpaired Student's *t* test). Source data are available online for this figure.

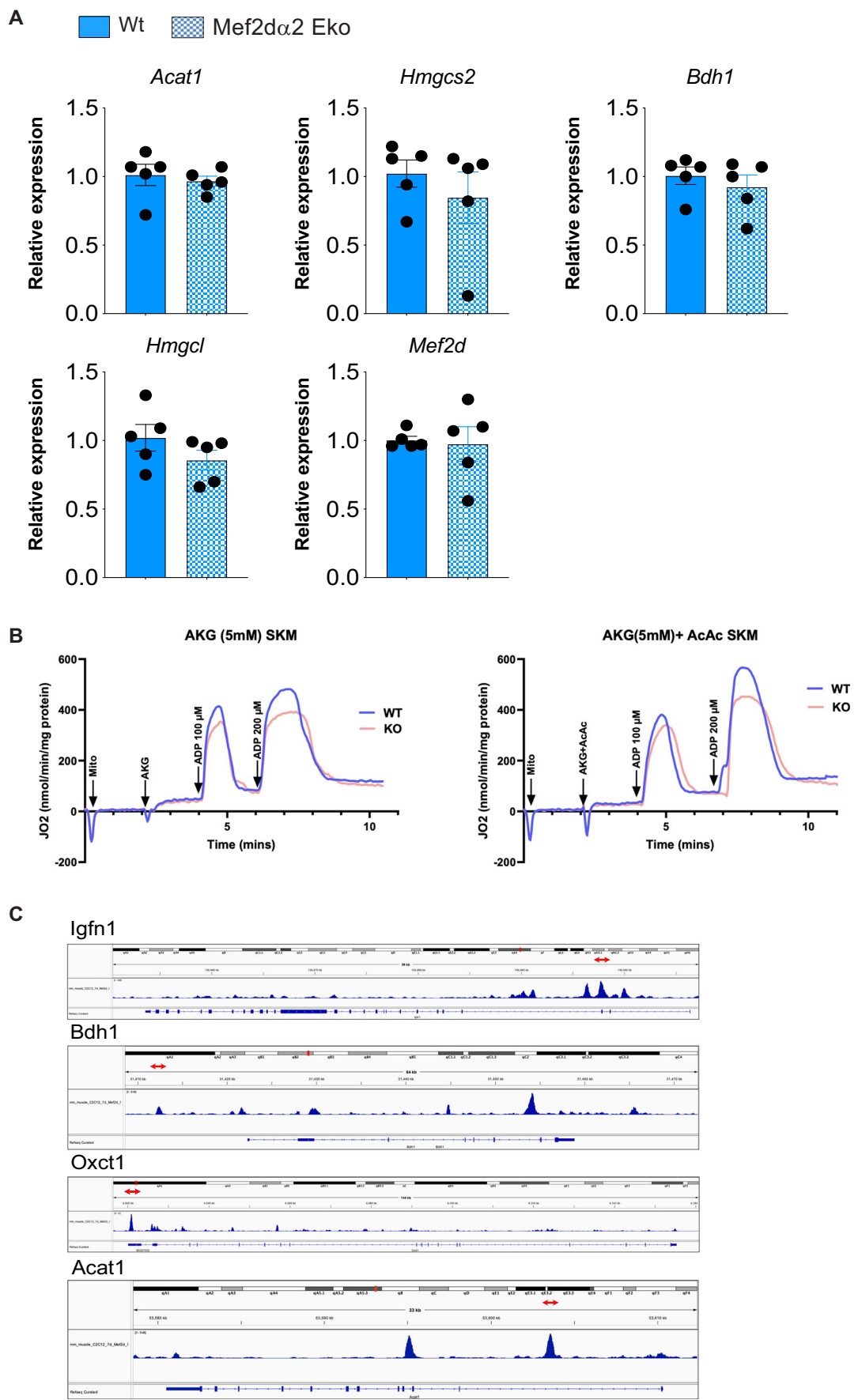

◀  **Figure EV5.  Gene expression related to ketone body metabolism does not differ in the livers of Mef2dα2 Eko mice.**

(A) RT-qPCR showing relative expression of indicated transcripts normalized to *Hprt* transcript levels using total RNA from WT and MEF2Dα2 Eko mice livers. Data are mean ± SEM, $n = 5$, Student's *t* test found no differences in genotypes. (B) Representative time-courses of isolated mitochondrial respiration for WT and MEF2Dα2 Eko mice transitioning from state 1 to state 4 respiration under AKG± AcAc. The respiratory rates are expressed as nmol/min/mg mitochondrial protein. The transitions from state 1 to state 4 respiration were monitored by first adding isolated mitochondria (0.05 mg/mL) to the respiration buffer at $t = 0$ min leading to state 1. At $t = 2$ min, substrates were added to energize the mitochondria, which led to state 2 respiration. This was followed by sequential additions of incremental ADP concentrations (100 and 200 μM). AKG: Alpha-ketoglutarate and AcAc: acetoacetate. (C) MEF2D ChIP data from Gönczi et al viewed on IGV viewer (version 2.19.4) for Igfn1, Bdh1, Oxct1, and Acat1 gene locus. The bi-directional arrowed line shows the region where we designed our primers for our analyses. Source data are available online for this figure.

