## [Peer Review File · EMBO Reports]

The muscle specific MEF2D 2 isoform promotes muscle ketolysis and running capacity in mice

Sushil Kumar, Xuan Ji, Hina Iqbal, Guan Xiangnan, Brittany Mis, Devanshi Dave, Suresh Kumar, Jacob Besler, Ranjan Dash, Zheng Xia, and Ravi Singh

Corresponding author(s): Ravi Singh (rksingh4@uh.edu) , Ranjan Dash (rdash@mcw.edu), Zheng Xia (xiaz@ohsu.edu)

Review Timeline:

Submission Date:	18th Sep 24
Editorial Decision:	30th Sep 24
Appeal Received:	24th Apr 25
Editorial Decision:	1st Jul 25
Revision Received:	15th Jul 25
Accepted:	25th Aug 25

Editor: Deniz Senyilmaz Tiebe

Transaction Report:

Dear Dr. Singh,

Thank you for submitting your manuscript to EMBO Reports. I have read your study carefully and discussed it with the other members of our editorial team including our chief editor Dr. Bernd Pulverer. I regret to inform you that we have decided not to pursue publication of this manuscript in its current form. However, we would be happy to reconsider the manuscript with some additional data as outlined below. In its current form of the manuscript, we recommend a transfer to Life Science Alliance should that be of interest.

I apologize for this unusual delay in getting back to you, which was caused by the current high rate of new submissions to our office, affecting our usually much shorter editorial handling time.

We appreciate your study demonstrating that depletion of specifically MEF2D α 2 splice isoform in mice leads to a decrease in running capacity and a reduction in ketone body utilization capacity. We realize that these findings are as such of interest to the field. However, we also conclude that as it stands, the study remains descriptive - i.e. in the absence of further molecular insight linking MEF2D α 2 to ketone body metabolism, the conceptual advance provided is not sufficient for EMBO Reports. As such, we are unable to publish your manuscript here in its current form. However, we would be happy to reconsider the manuscript in the presence of additional molecular insight by following strategies like a ChIP analysis checking whether MEF2D α 2 directly regulates ketolytic gene expression, or exploring whether MEF2D α 2 expression is regulated by exercise state, or a rescue experiment by manipulation of ketone body metabolism. I would be happy to discuss this aspect further should you wish.

That being said, and as mentioned above, your work in its current form is an excellent candidate for our partner journal Life Science Alliance (<http://www.life-science-alliance.org/>; our broad scope Open Access journal published in partnership between the EMBO-, Rockefeller University-, and Cold Spring Harbor Laboratory Presses). The editors of Life Science Alliance would be pleased to send your manuscript for in-depth peer review; no reformatting is required. We very much hope you will be interested in this option: please follow the link below for transfer. Eric Sawey, Executive Editor of Life Science Alliance (e.sawey@life-science-alliance.org), will be pleased to answer any questions.

I very much hope that you are interested in this option - please use the following link for transfer; no reformatting is required.

Yours sincerely,

Deniz Senyilmaz Tiebe, PhD
Senior Scientific Editor
EMBO Reports

** As a service to authors, EMBO Press provides authors with the ability to transfer a manuscript that one journal cannot offer to publish to another journal, without the author having to upload the manuscript data again. To transfer your manuscript to another EMBO Press journal using this service, please click on Link Not Available

Ravi K. Singh, Ph.D., Assistant Professor
Department of Pharmacological and Pharmaceutical Sciences,
College of Pharmacy, University of Houston
Health Building 2, Room 5023, 4349 Martin Luther King Boulevard
Houston, TX 77204-1217, Phone: 713-743-9770
Email: rksingh4@Central.UH.EDU
Website: <https://www.uh.edu/pharmacy/directory-home/pps-faculty/ravi-singh/index.php>

Dear Editor:

We are pleased to re-submit the manuscript, “**The muscle-specific MEF2D α 2 isoform promotes muscle ketolysis and running capacity in mice**”, to be considered as a research article. I discussed our research findings with new data via a Zoom call with Dr. Deniz Tiebe and she recommended re-submitting the manuscript to EMBO Reports.

Hundreds of muscle genes are alternatively spliced after birth with implication in muscle contractility, metabolism and exercise response, but the function of very few resultant protein isoforms is known. In this manuscript, we demonstrate that one muscle-specific MEF2D protein isoform, generated by alternative splicing, promotes muscle ketone body oxidation and running capacity, and systemic availability of ketone bodies in mice. I want to emphasize that genetic mouse knockout studies often show no phenotype, but in our study, we are not deleting the whole gene but simply deleting a muscle-specific isoform (the fetal isoform of the gene is still expressed) and observe a phenotype not only in skeletal muscle but in the whole body. Our findings are highly significant and point to new mechanisms regulating exercise capacity and future research to curb ketoacidosis in diabetic patients, which, if untreated, is lethal.

Ketone bodies are alternate fuel source generated by the liver in response to low carbohydrate availability in neonates and upon starvation and exhausting exercise. However, the utilization of the ketone body by peripheral organs is typically assumed to depend on substrate availability. Here, we discovered that the muscle-specific MEF2D α 2 protein isoform, which is generated by regulated postnatal alternative splicing, promotes the expression of all three ketolytic genes, Bdh1, Oxct1, and Acat1. Consistently, mice lacking the MEF2D α 2 isoform displayed a slower clearance of blood ketone body in a tolerance test, performed significantly less in a treadmill test, and displayed a higher level of ketone bodies after feeding a ketogenic diet. We utilized a high-resolution respirometer to demonstrate reduced utilization of ketone bodies in mitochondria isolated from muscles of mice lacking the MEF2D α 2 protein isoform. Lastly, we showed direct binding of MEF2D to genomic loci of all three ketolytic genes. Thus, we have discovered that MEF2D α 2 protein isoform generated by alternative splicing promotes skeletal muscle ketone body oxidation and running capacity in mice.

There are four MEF2 paralogous genes (Mef2a-d) in mammals, which are expressed in different tissues, and all paralogs undergo splicing to generate multiple isoforms in a developmentally regulated and tissue-specific manner. Mef2d undergoes robust postnatal and muscle-specific alternative splicing, resulting in the predominant adult muscle isoform called MEF2D α 2. Previous studies conducted by ourselves and other researchers have demonstrated the importance of this muscle-specific MEF2D α 2 isoform in muscle regeneration and myoblast fusion through overexpression experiments [1-3]. While it is well-known that MEF2 transcription factors play a crucial role in embryonic muscle development, their significance in adult skeletal muscle remains unclear. Moreover, the MEF2 transcriptional activity is upregulated after exercise, but the precise paralog or splice isoform important for determining the

exercise capacity or the adaptive response to exercise remains a mystery. Our work highlights the in vivo role of the highly conserved muscle-specific MEF2D α 2 isoform in skeletal muscle homeostasis and its impact on exercise performance. MEF2 transcription factors have been extensively studied for ~30 years, but our studies are first to identify a role of MEF2D in ketone body metabolism.

We believe our results will be relevant in humans as the muscle-specific inclusion of this exon is conserved in humans, and the Mef2 α 2 exon is skipped in the muscles of myotonic dystrophy patients [4]. Scientists working in the fields of RNA processing, muscle metabolism, and myotonic dystrophy will appreciate our novel findings and utilize them as a foundation for follow-up future investigations. It has been shown that supplementing with ketone esters before exercise enhances muscle performance [5], and our results suggest new insight into mechanisms allowing efficient utilization of the ketone bodies during the postnatal period and exercise. Hopefully, in the future, our discovery can also be utilized to modulate systemic ketone body levels by adjusting its use in skeletal muscle, i.e., to increase exercise capacity or to reduce ketone body levels for the prevention of diabetic ketoacidosis.

We thank you for consideration of this manuscript.

Sincerely,

Ravi K. Singh, Ph.D.
713-743-9770.

References:

1. Singh, R.K., Xia, Z., Bland, C.S., Kalsotra, A., Scavuzzo, M.A., Curk, T., Ule, J., Li, W., Cooper, T.A. Rbfox2-coordinated alternative splicing of Mef2d and Rock2 controls myoblast fusion during myogenesis(2014) Molecular Cell, 55 (4), pp. 592-603. Cited 95 times.
<https://www.scopus.com/inward/record.uri?eid=2-s2.0-84906791713&doi=10.1016%2fj.molcel.2014.06.035&partnerID=40&md5=aaeb1b2ff60eed1c479755549d034931>
2. Singh, R.K., Kolonin, A.M., Fiorotto, M.L., Cooper, T.A. Rbfox-Splicing Factors Maintain Skeletal Muscle Mass by Regulating Calpain3 and Proteostasis (2018) Cell Reports, 24 (1), pp. 197-208. Cited 37 times.
<https://www.scopus.com/inward/record.uri?eid=2-s2.0-85048989217&doi=10.1016%2fj.celrep.2018.06.017&partnerID=40&md5=f388df0c5ae78342b6b6e614c7b88026>

3. Sebastian, S., Faralli, H., Yao, Z., Rakopoulos, P., Pali, C., Cao, Y., Singh, K., Liu, Q.-C., Chu, A., Aziz, A., Brand, M., Tapscott, S.J., Dilworth, F.J. Tissue-specific splicing of a ubiquitously expressed transcription factor is essential for muscle differentiation
(2013) *Genes and Development*, 27 (11), pp. 1247-1259. Cited 94 times.
<https://www.scopus.com/inward/record.uri?eid=2-s2.0-84878849371&doi=10.1101%2fgad.215400.113&partnerID=40&md5=60b584cc2995b9f69c95eaf9094fbefe>
4. Thomas, J.D., Sznajder, Ł.J., Bardhi, O., Aslam, F.N., Anastasiadis, Z.P., Scotti, M.M., Nishino, I., Nakamori, M., Wang, E.T., Swanson, M.S.
Disrupted prenatal RNA processing and myogenesis in congenital myotonic dystrophy
(2017) *Genes and Development*, 31 (11), pp. 1122-1133. Cited 70 times.
<https://www.scopus.com/inward/record.uri?eid=2-s2.0-85025064559&doi=10.1101%2fgad.300590.117&partnerID=40&md5=fe99ab395ebb54387580821882ff5c69>
5. Cox, P.J., Kirk, T., Ashmore, T., Willerton, K., Evans, R., Smith, A., Murray, A.J., Stubbs, B., West, J., McLure, S.W., King, M.T., Dodd, M.S., Holloway, C., Neubauer, S., Drawer, S., Veech, R.L., Griffin, J.L., Clarke, K. Nutritional Ketosis Alters Fuel Preference and Thereby Endurance Performance in Athletes
(2016) *Cell Metabolism*, 24 (2), pp. 256-268. Cited 395 times.
<https://www.scopus.com/inward/record.uri?eid=2-s2.0-84991080555&doi=10.1016%2fj.cmet.2016.07.010&partnerID=40&md5=0515cb17dda7424145a2bdd71fadd969>

Dear Ravi,

Thank you for submitting your manuscript to EMBO Reports, which was now seen by two referees, whose reports are copied below.

Referees express interest in the proposed role of the muscle-specific MEF2D α 2 isoform in regulation of muscle ketolysis and running capacity in mice. As you will see, referees find that the study of interest and support publication with minor changes.

Moreover, the editorial points below need to be addressed before I can accept the manuscript.

- Please address the minor concerns of referee #2.
- Please provide 3-5 keywords for your study. These will be visible in the html version of the paper and on PubMed and will help increase the discoverability of your work.
- As per our guidelines, please add a 'Data Availability Section', where datasets and computer code that were generated in the reported study should be listed in a structured manner. Each dataset should be listed under a separate bullet point that includes 1) a short description of the measurement type (eg RNA-Seq, ChIP-Seq, mass spectrometry proteomics, imaging, etc...), 2) the name of the repository (or its recommended acronym, see table below and consult fairsharing.org); 3) the DOI or accession number of the dataset; and 4) a resolvable link to the dataset, either in the form of a resolvable link from <http://identifiers.org> or as the full URL to the respective database record. If your study does not include datasets, please insert the following statement only: This study includes no data deposited in external repositories.
- Please add a "Disclosure and Competing Interests Statement" section (<https://www.embopress.org/page/journal/14693178/authorguide#conflictsofinterest>).
- Please remove the "Author contribution" section from the manuscript.
- As per our format requirements, in the reference list, citations should be listed in alphabetical order and then chronologically, with the authors' surnames and initials inverted; where there are more than 10 authors on a paper, 10 will be listed, followed by 'et al.'. Please see <https://www.embopress.org/page/journal/14693178/authorguide#referencesformat>
- We note the usage of the phrase "data not shown" on pages 5, 6 and 11, which is not allowed as per journal policy. Please either show the data or remove the relevant conclusion.
- Please fill out and include an author checklist as listed in our online guidelines (<https://www.embopress.org/page/journal/14693178/authorguide>)
- We note that the funding information is not congruent in the manuscript tracking system and the manuscript text - the grant number for the American Heart Association is not the same (15SDG25610021 in the manuscript vs. 4SDG20450053 in the manuscript tracking system); the following are missing from the manuscript tracking system: funds from the Medical College of Wisconsin (MCW) and University of Houston UH), pilot funding from the Research Affairs Committee (MCW) and Children's Wisconsin Research Institute.
- Currently, all figures (main and suppl.) provided in one PDF. For publication, we need individual production quality Figure files for the main figures. Figs S1 to S5 should be converted to EV figures, they also need to be provided separately and the nomenclature in all places should be Figure EV1, etc. the legends should stay in the manuscripts. For publication, we require TIFF, PDF or EPS files in PC or Macintosh format, preferably from PhotoShop or Illustrator software. For any figures submitted in Photoshop or TIF(F) format we require layered files to be sent whereby all text, arrows or additional attributes are placed on individual layers within the file. For line art/charts/graphs we prefer to work with Adobe Illustrator AI, EPS, or high-resolution PDF files.
- We note that Tables S1 and S2 called out but they are missing.
- All research articles submitted as revised versions must include a structured methods section that includes a Reagents and Tools Table followed by a Methods and Protocols section. Please see <https://www.embopress.org/page/journal/14693178/authorguide#structuredmethods> for further information.
- The manuscript sections should be in the following order: Title page - Abstract & Keywords - Introduction - Results - Discussion - Methods - Data Availability - Acknowledgments - Disclosure Statement & Competing Interests - References - Figure Legends - (Main Tables with legends if applicable) - Expanded View Figure Legends.
- Experimental Procedures should be renamed as Methods.
- Our production/data editors have asked you to clarify several points in the figure legends - Figure Legends (main + EV):
 - o Please note that figure titles are not provided for figures 1-5, S1-S3, S5. Kindly rectify the same.
 - o Please note that the sub-figure S4 E is mislabeled as sub-figure S4 D in the manuscript. This needs to be rectified."
 - o Please note that the exact p values are not provided in the legends of figures 2A, B; 3C, D, E, G; 4C, D; 5A-E; S1 E, S2 A, B; S3 C, D, E, F, G, H; S4 C-E
 - o Please indicate the statistical test used for data analysis in the legends of figures S4 A, B
 - o Please note that information related to n is missing in the legends of figures 3D, 4D, 5B, S4 D, S5 A
 - o Please note that the error bars are not defined in the legends of figures 3D, S4 D, S5 A
 - o Please note that scale bar and its definition are missing for figure 2D
- Papers published in EMBO Reports include a 'synopsis' and 'bullet points' to further enhance discoverability. Both are displayed on the html version of the paper and are freely accessible to all readers. The synopsis includes a short standfirst summarizing the study in 1 or 2 sentences (max 35 words) that summarize the paper and are provided by the authors and

streamlined by the handling editor. I would therefore ask you to include your synopsis blurb and 3-5 bullet points listing the key experimental findings.

• In addition, please provide an image for the synopsis. This image should provide a rapid overview of the question addressed in the study but still needs to be kept fairly modest since the image size cannot exceed 550 (width) x 300-600 (height) pixels.

Thank you again for giving us to consider your manuscript for EMBO Reports, I look forward to your minor revision.

Kind regards,

Deniz

--

Deniz Senyilmaz Tiebe, PhD
Senior Scientific Editor
EMBO Reports

Referee #1:

The paper investigates the role of alternatively spliced isoforms of Mef2d in skeletal muscle metabolism. This is an important investigation; while the functions of different Mef2 paralogs have been thoroughly investigated, much less is known regarding the roles of the conserved tissue specific and developmentally regulated isoforms. The approach used genomic deletion of the Mef2d α 2-exon to test the function of this isoform in adult animals by its absence. It is interesting that the mice showed no structural changes (muscle weight, fiber type, fiber number) but showed physiological effects (reduced running capacity) which indicates an intrinsic defect in muscle, rather than simply gross tissue damage causing a phenotype. The results of RNA-sequencing revealed a signature consistent with altered ketone body utilization that was demonstrated experimentally by systematic analysis.

The authors performed a thorough characterization of structural and metabolic effects and showed no difference in strength, insulin resistance, glycogen levels in sedentary and exercised mice, blood lactate levels all of which showed no differences. The authors present a strong case using biochemical and transcriptomic analysis that the Mef2d α 2 isoform, the predominant isoform in adult skeletal muscle, is a major player in utilization of ketone bodies. This is a novel finding in and of itself. The mechanism appears to be direct as Mef2d α 2 was shown by the authors to bind to ketolytic genes. The results are convincing as the experiments were performed systematically and rigorously including use of two independent founder lines and backcrossing the CRISPR/Cas9-derived mice to essentially eliminate off target effects.

Referee #2:

The authors report investigation of the function of MEF2D α 2, a muscle-specific postnatal alternative splicing isoform of MEF2D. MEF2D α 2 exon knockout (Eko) mouse was created using CRISPR/Cas9 and characterized for muscle function. While sedentary MEF2D α 2 Eko mice did not display any notable phenotype, they had reduced running capacity and reduced capacity for muscle ketolysis after exercise or on a ketogenic diet. The authors further showed that MEF2D α 2 most likely regulates the expression of ketolytic genes. Overall, the work is rigorous (e.g., two independent mouse lines were characterized), the data are of high quality, and the conclusions are supported by the experimental evidence. The findings are novel and of significance in the field.

Minor points:

1. Fig. 3B: the x-axis should be labeled - time? unit?
2. In Discussion, "Fig. 5s" should be "Fig. 5a".

Dear Deniz,

Thank you for your pre-accept decision and for overseeing the final review of our manuscript. We also thank the reviewers and editorial team for their timely and careful review of our application and for helpful suggestions. The revised version now corrects the two minor mistakes pointed out by the reviewer. In addition, we have updated the abstract so that it reads better. Below is our point-by-point response to the reviewer's and editorial comments.

We hope that the manuscript is now appropriate for publication in *EMBO Reports*.

Sincerely,

Ravi Singh

• Please address the minor concerns of referee #2.

1. Fig. 3B: the x-axis should be labeled - time? unit?

X-axis is now labeled in the revised figure.

2. In Discussion, "Fig. 5s" should be "Fig. 5a".

The mistake has been corrected in the discussion section.

• Please provide 3-5 keywords for your study. These will be visible in the html version of the paper and on PubMed and will help increase the discoverability of your work.

The keywords are not included in the revised version on the title page.

• As per our guidelines, please add a 'Data Availability Section', where datasets and computer code that were generated in the reported study should be listed in a structured manner. Each dataset should be listed under a separate bullet point that includes

1) a short description of the measurement type (eg RNA-Seq, ChIP-Seq, mass spectrometry proteomics, imaging, etc...),

2) the name of the repository (or its recommended acronym, see table below and consult fairsharing.org);

3) the DOI or accession number of the dataset; and

4) a resolvable link to the dataset, either in the form of a resolvable link from <http://identifiers.org> or as the full URL to the respective database record. If your study does not include datasets, please insert the following statement only: This study includes no data deposited in external repositories.

The data availability section is now included in the revised version. The RNA-seq data has been deposited to GEO under the accession number GSE302518. The accession number is hyperlinked with the URL (see below)

URL: <https://www.ncbi.nlm.nih.gov/geo/query/acc.cgi?acc=GSE302518>

- Please add a "Disclosure and Competing Interests Statement" section (<https://www.embopress.org/page/journal/14693178/authorguide#conflictsofinterest>).

This section is now included.

- Please remove the "Author contribution" section from the manuscript.

Removed.

- As per our format requirements, in the reference list, citations should be listed in alphabetical order and then chronologically, with the authors' surnames and initials inverted; where there are more than 10 authors on a paper, 10 will be listed, followed by 'et al.'. Please see <https://www.embopress.org/page/journal/14693178/authorguide#referencesformat>

The manuscript is formatted as requested.

- We note the usage of the phrase "data not shown" on pages 5, 6 and 11, which is not allowed as per journal policy. Please either show the data or remove the relevant conclusion.

"Data not shown" is removed. It doesn't affect the conclusions of our results.

- Please fill out and include an author checklist as listed in our online guidelines (<https://www.embopress.org/page/journal/14693178/authorguide>)

Checklist is now completed and uploaded.

- We note that the funding information is not congruent in the manuscript tracking system and the manuscript text - the grant number for the American Heart Association is not the same (15SDG25610021 in the manuscript vs. 4SDG20450053 in the manuscript tracking system); the following are missing from the manuscript tracking system: funds from the Medical College of Wisconsin (MCW) and University of Houston UH), pilot

funding from the Research Affairs Committee (MCW) and Children's Wisconsin Research Institute.

15SDG25610021 is the correct number. All other funding sources (MCW, Children's Wisconsin, and UH) are now included in the manuscript tracking system.

- Currently, all figures (main and suppl.) provided in one PDF. For publication, we need individual production quality Figure files for the main figures. Figs S1 to S5 should be converted to EV figures, they also need to be provided separately and the nomenclature in all places should be Figure EV1, etc. the legends should stay in the manuscripts. For publication, we require TIFF, PDF or EPS files in PC or Macintosh format, preferably from PhotoShop or Illustrator software. For any figures submitted in Photoshop or TIF(F) format we require layered files to be sent whereby all text, arrows or additional attributes are placed on individual layers within the file. For line art/charts/graphs we prefer to work with Adobe Illustrator AI, EPS, or high-resolution PDF files.

I'm using Affinity Designer to prepare the figures. The high-resolution PDF files for all the figures are uploaded.

- We note that Tables S1 and S2 called out but they are missing.

Table EV1 (stats of RNA-seq data alignment), EV2 (differentially expressed genes) and EV3 (RT-qPCR primer list) are now added.

- All research articles submitted as revised versions must include a structured methods section that includes a Reagents and Tools Table followed by a Methods and Protocols section. Please see <https://www.embopress.org/page/journal/14693178/authorguide#structuredmethods> for further information.

Reagents_Tools_Table file is now added.

- The manuscript sections should be in the following order: Title page - Abstract & Keywords - Introduction - Results - Discussion - Methods - Data Availability - Acknowledgments - Disclosure Statement & Competing Interests - References - Figure Legends - (Main Tables with legends if applicable) - Expanded View Figure Legends.

Manuscript arranged in the suggested order except methods section (see the response to the question above).

- Experimental Procedures should be renamed as Methods.

Done.

• Our production/data editors have asked you to clarify several points in the figure legends - Figure Legends (main + EV):

o Please note that figure titles are not provided for figures 1-5, S1-S3, S5. Kindly rectify the same.

Figure titles are now added.

o Please note that the sub-figure S4 E is mislabeled as sub-figure S4 D in the manuscript. This needs to be rectified."

Corrected.

o Please note that the exact p values are not provided in the legends of figures 2A, B; 3C, D, E, G; 4C, D; 5A-E; S1 E, S2 A, B; S3 C, D, E, F, G, H; S4 C-E

The exact pValues are not added either in the figure or in the figure legend. A one-way ANOVA with multiple comparisons between 4 groups has six different p-values, which we have included in the main figure, which makes the figure look a little cluttered. It will be difficult to put these pValues in the figure legend in a way that is easily interpreted by the reader.

o Please indicate the statistical test used for data analysis in the legends of figures S4 A, B

Statistical test used is now indicated in the figure legend.

o Please note that information related to n is missing in the legends of figures 3D, 4D, 5B, S4 D, S5 A

N is now indicated in the figure legend.

o Please note that the error bars are not defined in the legends of figures 3D, S4 D, S5 A
Error bars now defined.

o Please note that scale bar and its definition are missing for figure 2D

Scale bar is now included.

• Papers published in EMBO Reports include a 'synopsis' and 'bullet points' to further

enhance discoverability. Both are displayed on the html version of the paper and are freely accessible to all readers. The synopsis includes a short standfirst summarizing the study in 1 or 2 sentences (max 35 words) that summarize the paper and are provided by the authors and streamlined by the handling editor. I would therefore ask you to include your synopsis blurb and 3-5 bullet points listing the key experimental findings.

A separate file with synopsis and bullet points are uploaded as related manuscript file.

• In addition, please provide an image for the synopsis. This image should provide a rapid overview of the question addressed in the study but still needs to be kept fairly modest since the image size cannot exceed 550 (width) x 300-600 (height) pixels.

Synopsis image is included.

Dr. Ravi Singh
University of Houston
Pharmacological and Pharmaceutical Sciences and Institute of Muscle Biology and Cachexia
4349 Martin Luther King Boulevard, Room 5023
Houston, TX 77204
United States

Dear Ravi,

Thank you for submitting your revised manuscript. I have now looked at everything and all is fine. Therefore, I am very pleased to accept your manuscript for publication in EMBO Reports.

Congratulations on a nice work!

Before we can export your manuscript to our production team, I need your input on two minor points:

1. Please add an URL into the Data Availability section, which directly resolves to GSE302518. I attached the manuscript file to this email.

2. Moreover, the synopsis image needs to be 550px wide and 300-600px high. When your synopsis image is resized accordingly, the labels are too small to read (please see attached). Please provide a synopsis image with larger labels.

You can send the manuscript text file back with the addition of the URL and the edited synopsis image by responding to this email. Thank you.

Kind regards,

Deniz
--
Deniz Senyilmaz Tiebe, PhD
Senior Scientific Editor
EMBO Reports
